# Active Learning with Foundation Model Priors: Efficient Learning under Class Imbalance

**Jiancheng Zhang** [1]  **Meiqing Li** [2]  **Qi Zhang** [3]  **Yinglun Zhu** [1]

## Abstract

Real-world datasets across image and text domains are often characterized by skewed class distributions and noisy annotations, which jointly degrade model performance, particularly on minority classes. Among existing solutions, active learning offers an effective and efficient paradigm by selectively querying the most informative and balanced samples for annotation. We propose an innovative active learning framework that mitigates class imbalance and selects the most informative samples to annotate. Leveraging foundation model priors, our algorithm enables imbalance-aware co-decisions between foundation model and small model to tackle noisy and imbalanced labels across various domains. We introduce the first study to systematically explore active learning under the dual challenges of label noise and class imbalance across image and text domains. Extensive experiments on imbalanced datasets demonstrate that our method achieves substantial annotation savings—over 50% compared to the best active learning baseline—while preserving performance and robustness to label noise.

## 1. Introduction

Modern deep learning models have demonstrated strong performance across a wide range of applications, but typically rely on large amounts of annotated training data. Active learning (Settles, 2009) aims to reduce annotation cost by sequentially and selectively querying the most informative samples for labeling. Despite its success (Gal et al., 2017; Sener & Savarese, 2017; Ash et al., 2019; Zhang et al., 2023a), applying active learning in real-world

settings introduces additional challenges, most notably class imbalance and label noise.

Under imbalanced data distributions, naively allocating labeling budgets often leads to oversampling majority classes, leaving insufficient labeled examples for rare classes and limiting effective model training. To address this issue, a line of recent work has proposed active learning strategies that promote class balance or improve minority-class coverage during sample selection (Aggarwal et al., 2020; Zhang et al., 2022; Nuggehalli et al.). In large-scale annotation scenarios, these difficulties are frequently compounded by label noise (Khosla et al., 2022), a common issue arising from annotator fatigue and perceptual inconsistencies.

Recent advances in foundation models, including language models (Achiam et al., 2023; Touvron et al., 2023; Liu et al., 2024) and vision models (Radford et al., 2021; Zhai et al., 2023), offer new opportunities for improving data efficiency. Owing to their rich prior knowledge, foundation models can provide informative signals for labeling and data selection, even in low-resource or imbalanced settings. Several recent studies combine active learning with foundation models (Bhatt et al., 2024; Xia et al., 2025; Gupte et al., 2024; Zhang & Zhu, 2025), using either standard active learning criteria or foundation model-driven query strategies to guide model training. However, to the best of our knowledge, foundation model-based active learning has not been systematically studied in the presence of both class imbalance and label noise.

In this paper, we propose a novel foundation model-based active learning algorithm for both class imbalance and label noise across image and text domains. Specifically, we first obtain predictions from the foundation model as a way to extract its encoded knowledge, which serve as prior information to guide the subsequent steps, rather than a Bayesian prior in the probabilistic inference sense. Then following the product of experts (PoE) introduced by Hinton (1999; 2002), we construct a combined probability as PoE in our setting to enable joint decision-making, where we integrate class-imbalance awareness to account for the challenges posed by imbalanced data. We then propose an imbalance-aware entropy filtering, which allows us to obtain a clean set with pseudo labels and a noisy set, some of whose samples have

---

[1]University of California, Riverside [2]Carnegie Mellon University [3]Worcester Polytechnic Institute. Correspondence to: Jiancheng Zhang <jzhan745@ucr.edu>, Yinglun Zhu <yzhu@ucr.edu>.

*Proceedings of the 43rd International Conference on Machine Learning*, Seoul, South Korea. PMLR 306, 2026. Copyright 2026 by the author(s).

true labels. This can exploit the complementary strengths of foundation model and active learning, while specifically addressing noise stemming from pseudo labels. During the final phase, the small model is fine-tuned by jointly leveraging guidance from the foundation model and curated data obtained through active learning. Whole pipeline implicitly mitigates the impact of oracle label noise, as evidenced by our empirical results.

**Our Contributions.** Our main contributions are as follows:

- To the best of our knowledge, we are among the first to address the non-trivial yet widespread scenario where both imbalance and label noise coexist in the text domain.

- We propose a novel active learning algorithm that leverages prior information from a foundation model to guide active learning through joint decision-making with a small model, while explicitly accounting for class imbalance.

- We perform experiments across 21 dataset settings, comprising both image and text domains. Our method consistently outperforms existing baselines by saving more than 50% annotation cost compared to the best algorithm.

**Paper Organization.** The rest of this paper is organized as follows. Section 2 introduces our problem formulation and presents our algorithm. In Section 3, we conduct extensive experiments to verify the effectiveness of our method and we provide additional analysis. Section 4 concludes the paper. Related work, implementation details, and supplementary experiments are deferred to the Appendix.

## 2. Methods

In this section, the background of the proposed method is introduced firstly in Section 2.1, then we provide our method, which consists of three main phases: prior labeling (Section 2.2); imbalance-aware uncertainty sampling and small model training (Section 2.3), as shown in Figure 1. A detailed description of the algorithm is provided in the Algorithm 1.

### 2.1. Preliminary

We study the pool-based active learning problem applicable to both text and image domains. Given the dataset $\mathcal{D}$, the initial unlabeled pool is denoted as $\mathcal{D}_U = \mathcal{D}$, where $\mathcal{D}_U = \{x_1, x_2, ..., x_N\}$, using a unified notation for both domains. Their labels $Y = \{y_1, y_2, ..., y_N\}$ are initially unknown. In this work, we study the multi-class classification problem,

where the corresponding ground-truth label $y$ belongs to the label space $\mathcal{Y} := [K]$, consisting of $K$ classes. In addition, we define the imbalance ration as $\gamma = \frac{\min_{k \in [K]} N_k}{\max_{k' \in [K]} N_{k'}}$, where $N_k$ denotes the number of examples in $\mathcal{D}_U$ of $k$-th class.

The active learning algorithm is executed over $T$ iterations. During the $t$-th iteration, the algorithm is provided with the labeled and unlabeled pools of examples, $\mathcal{D}_L^t$ and $\mathcal{D}_U^t$ respectively, where $\mathcal{D}_L^t \cup \mathcal{D}_U^t = \mathcal{D}$ and $\mathcal{D}_L^t \cap \mathcal{D}_U^t = \emptyset$. The active learning method then selects $B$ examples from the unlabeled pool $\mathcal{D}^t \subseteq \mathcal{D}_U^t$ and subsequently queries their corresponding labels from the oracle $O$. Then the labeled and unlabeled pools are updated: $\mathcal{D}_L^{t+1} \leftarrow \mathcal{D}_L^t \cup \mathcal{D}^t$, $\mathcal{D}_U^{t+1} \leftarrow \mathcal{D}_U^t \setminus \mathcal{D}^t$. Based on the new labeled pool $\mathcal{D}_L^{t+1}$ and corresponding labels, the model $f_t$ is trained to guide the selection in the next iteration. Whole active learning algorithm aims to achieve high accuracy with the least possible annotation cost.

### 2.2. Prior Labeling

Prior labeling aims to extract the predictive outputs from the foundation model and small model, respectively. Specifically, both models generate predicted probabilities for the unlabeled pool $\mathcal{D}_U$, and pseudo labels derived from the foundation model's predictions—together with these probabilities—are passed to the next phase. The key insight is that the decisions of the foundation model and small models jointly influence the subsequent data selection process.

First, foundation model generates the probabilities $p_L(x, y = i), \forall i \in [K]$ for each example in the unlabeled pool. Based on the predicted probabilities from the foundation model, the corresponding pseudo labels are obtained as:

$$\bar{y} = \arg \max_{i \in [K]} p_L(x, y = i). \tag{1}$$

Since both models' predictions are used in the subsequent process, the foundation model's probabilities $p_L$ are preserved after generating the pseudo labels. We then obtain the small model's predictions from the previous iteration, $p_f(x, y = i)$, in a manner similar to that of the foundation model. Both models' predicted probabilities, $p_L$ and $p_f$, along with the pseudo labels $\bar{y}$ derived from the foundation model, are forwarded to the subsequent phase.

### 2.3. Imbalance-aware Uncertainty Sampling for Small Model Training

Before training the small model, we perform the phase 2, as illustrated in Figure 1, which constructs the training pool—comprising the labeled pool and a clean set—by combining an imbalance-aware entropy filtering based on PoE with uncertainty-based sampling.

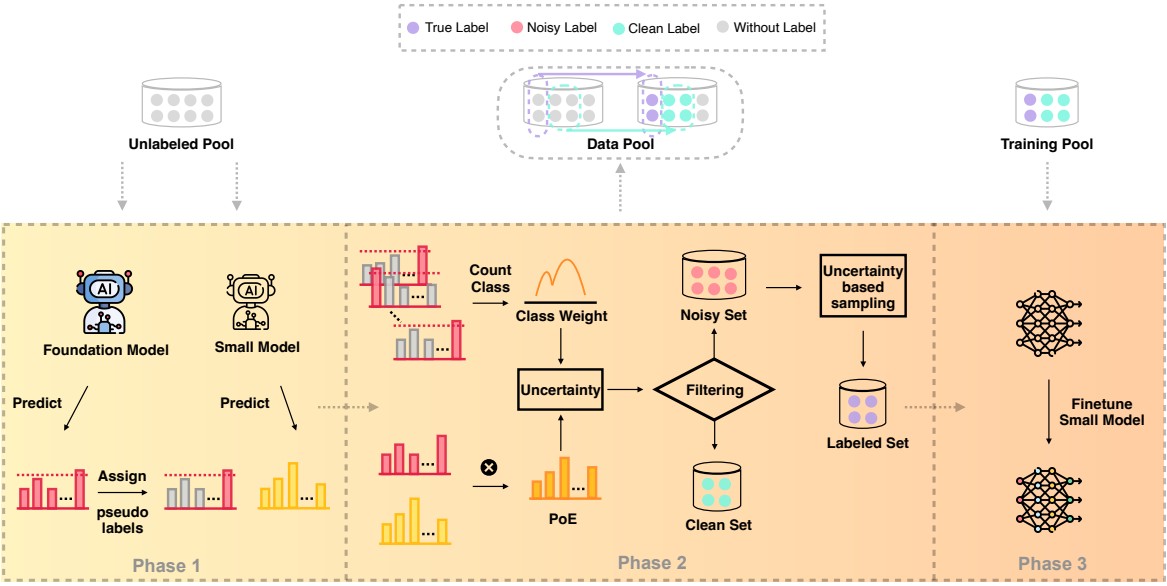

*Figure 1.* The overview of our framework. The framework consists of three main phases: prior labeling, uncertainty-based sampling with imbalance awareness, and small model training, detailed in Section 2. Given the unlabeled pool, **first phase** generates the predicted probabilities from the foundation model and small model, while pseudo labels are derived from the foundation model's predictions. In the **second phase**, the clean and noisy sets are constructed based on the imbalance-aware entropy (uncertainty) of the products of experts (PoE). Then labeled set is generated by uncertainty sampling. Lastly, small model is trained to inform the next iteration.

**Products of Experts (PoE).** To perform imbalance-aware entropy filtering, we firstly need to construct the PoE. Inspired by the prior works Hinton (1999; 2002) that introduce PoE to satisfy different constraints by multiplying the probabilities from different models, we define $\bar{p}$ as the PoE in our work to allow the decisions of both the foundation model and the small model to jointly influence the subsequent data selection process, computed as:

$$\bar{p}(x, y = i) = \frac{p_f(x, y = i) \cdot p_L(x, y = i)}{\sum_j p_f(x, y = j) \cdot p_L(x, y = j)}. \quad (2)$$

where $p_f$ is the probabilities from a small model and $p_L$ is the probabilities from a pretrained foundation model. The PoE jointly accounts for the Pretrained Priors and the capacity of the small model.

**Imbalance-aware Entropy Filtering.** While accounting for class imbalance, and aiming to select training data without incurring annotation costs, we propose an imbalance-aware entropy filtering, which can produce a clean set that forms the subsequent training pool for small model training.

Based on the PoE, we compute an entropy to jointly measure the uncertainty of both the foundation model and small model over the unlabeled pool, which is calculated as follows:

$$H(x) = -\sum_i \bar{p}(x, y = i) \log \bar{p}(x, y = i), \quad (3)$$

which can be used to filter the unlabeled pool. Since it ignores class imbalance, we extend it with an imbalance-aware modification. Inspired by the prior works on class imbalance (Cui et al., 2019; Zhang et al., 2023b), we propose an imbalance-aware boosted entropy that can weight the naive entropy according to class imbalance. In details, we define the $w_i$ as the imbalance-aware weight for class $i$, which is computed as the:

$$w_i = |\{x \in \mathcal{D}_U^t : \bar{y}(x) = i\}|. \quad (4)$$

By incorporating this weight, the entropy accounts for class imbalance when measuring the uncertainties of both models. Following normalization, we define the imbalance-aware boosted entropy as follows:

$$H_b(x) = -\sum_i (\bar{p}(x, y = i) \log \bar{p}(x, y = i) \cdot \frac{w_{\bar{y}(x)}}{w_{\max}}), \quad (5)$$

where $w_{\max} = \max_i w_i$. If the entropy $H_b(x)$ is very large indicating low confidence from both the foundation model and small model, the noisy set can be derived as follows:

$$\mathcal{D}_N := \arg \max_{\mathcal{S} \subseteq \mathcal{D}_U^t, |\mathcal{S}| = \rho |\mathcal{D}_U^t|} \sum_{x \in \mathcal{S}} H_b(x), \quad (6)$$

where $\rho$ is a predefined hyperparameter that controls the size of the noisy set. Otherwise, the remaining examples together with their corresponding pseudo labels form the clean set $\mathcal{D}_C$, reflecting high confidence from both models:

$$\mathcal{D}_C := \{(x, \bar{y}) \mid x \in \mathcal{D}_U^t \setminus \mathcal{D}_N\}, \quad (7)$$

---

**Algorithm 1** PriorAL: Foundation Model Priors-Informed Active Learning under Class Imbalance

---

**Input:** Pretrained foundation model $M_L$, small model $f$, human annotation oracle $O$, human annotation per-round budget $B$, imbalanced dataset $\mathcal{D}$, and percentage $\rho$.

1: Initialize labeled pool $\mathcal{D}_L^1 \leftarrow M$ examples drawn uniformly at random from $\mathcal{D}$ together with queried labels and unlabeled pool $\mathcal{D}_U^1 = \mathcal{D} \setminus \mathcal{D}_L^1$.
2: Train small model $f$ with $\mathcal{D}_L^1$ to get $f_0$.
3: **for** $t = 1, 2 \ldots, \text{T}$ **do**
4:     **Phase 1: Prior Labeling.**
5:     Use foundation model $M_L$ to generate predicted probabilities $p_L(x, y = i)$ for each data point in the unlabeled pool $x \in \mathcal{D}_U^t$, and derive its pseudo label $\bar{y}$ accordingly.
6:     We use small model from the last iteration $f_{t-1}$ to obtain probabilities $p_f(x, y = i)$.
7:     **Phase 2: Imbalance-aware Uncertainty Sampling.**
8:     Compute the products of experts $p\prime(x, y = i) = p_f(x, y = i) \cdot p_L(x, y = i)$ and normalize the new probabilities $\bar{p}(x, y = i) = \frac{p\prime(x, y=i)}{\sum_j p\prime(x, y=j)}$.
9:     Calculate the imbalance-aware boosted entropy for all $x \in \mathcal{D}_U^t$:

$$H_b(x) = -\sum_i (\bar{p}(x, y = i) \log \bar{p}(x, y = i) \cdot \frac{w_{\bar{y}(x)}}{w_{\max}}),$$

    where $\bar{y}(x) = \arg\max_i p_L(x, y = i)$, $w_i = |\{x \in \mathcal{D}_U^t : \bar{y}(x) = i\}|$, and $w_{\max} = \max_i w_i$.
10:   Set the noisy set $\mathcal{D}_N$ as the top-$\rho|\mathcal{D}_U^t|$ elements of $\mathcal{D}_U^t$ with the largest $H_b(x)$ as Eq.(6), and clean set $\mathcal{D}_C$ as Eq.(7).
11:   We use small model from the last iteration $f_{t-1}$ to obtain probabilities $p_f(x, y = i)$, for each $x \in \mathcal{D}_N$. Compute uncertainty scores $U_f(x) = -\sum_i p_f(x, y = i) \log p_f(x, y = i)$.
12:   Ask annotation oracle $O$ to label the top-$B$ samples in $\mathcal{D}_N$ with the largest uncertainty scores $U_f(x)$, indicating the highest uncertainty.
13:   Denote this set of $B$ data as $\mathcal{D}_t$, and update $\mathcal{D}_L^{t+1} = \mathcal{D}_L^t \cup \mathcal{D}_t$ and $\mathcal{D}_U^{t+1} = \mathcal{D}_U^t \setminus \mathcal{D}_t$.
14:   **Phase 3: Small Model Training.**
15:   Train $f_{t-1}$ with data points in $\mathcal{D}_C \cup \mathcal{D}_L^{t+1}$ to get $f_t$.
16: **end for**

---

where $\bar{y}$ is the pseudo label predicted by the foundation model, as described in the Section 2.2. The filtered clean set is incorporated into the training pool and used to train the small model in the final phase. It is worth noting that the clean set is produced without incurring any annotation cost, but can nevertheless be directly leveraged for training the small model. For the noisy set, due to their low confidence and unreliable pseudo labels, these instances are sent to the oracle to obtain the ground-truth annotations.

**Uncertainty-based Sampling.** To maximize the efficiency of limited annotation cost, we impose an uncertainty-based sampling strategy on the noisy set. We begin by generating the class probabilities $p_f(x, y = i), \forall x \in \mathcal{D}_N$ on the noisy set with the small model from the last iteration. To quantify the uncertainty of the small model, we define $U_f(x)$ as its uncertainty score, which is computed as:

$$U_f(x) = -\sum_i p_f(x, y = i) \log p_f(x, y = i). \quad (8)$$

This uncertainty metric follows the similar formulation as the entropy described earlier. We select the top-$B$ samples with the highest uncertainty scores, indicating low confidence, from $\mathcal{D}_N$ and query the oracle for their annotations. We denote the newly obtained set as $\mathcal{D}_t$, which is then used to update the labeled and unlabeled data pools.

**Small Model Training.** To achieve high predictive accuracy for the trained small model while annotating as few examples as possible, we construct the training pool, which consists of the labeled set and the clean set obtained in phase 2. The key distinction is that the labeled set contains ground-truth annotations, whereas the clean set is assigned high-confidence pseudo labels generated by the foundation model. In the final phase of each iteration, the small model is trained using the constructed training pool, and the updated model is then passed to the first phase of the next iteration.

## 3. Experiments

### 3.1. Setup

**Datasets.** To evaluate the performance of our proposed method in both text and image domains, we perform ex-

*Table 1.* Performance of our method over existing baselines for both merge imbalance and long-tailed imbalance settings. We report the **balanced pool accuracy** and **balanced test accuracy** evaluated under 10% and 20% annotation cost across text (Trec, AGNews and SST-2 datasets) and image (CIFAR-10, CIFAR-100 and PathMNIST datasets) domains. Bold numbers denote the best performance for each dataset and second-best results are underlined.

| | Merge Imbalance | | | | | | | | Long-tailed Imbalance | | | | | | | | | | | |
| | Text Domain | | | | Image Domain | | | | Text Domain | | | | | | Image Domain | | | | | |
| Method | Trec | | AGNews | | C10 | | C100 | | Trec | | AGNews | | SST-2 | | C10 | | C100 | | PathMNIST | |
|---|---|---|---|---|---|---|---|---|---|---|---|---|---|---|---|---|---|---|---|---|
| **Balanced Pool Accuracy** | | | | | | | | | | | | | | | | | | | | |
| Random | 82.39 | 88.78 | 75.06 | 84.56 | 71.52 | 77.61 | 43.50 | 53.73 | 83.68 | 90.74 | 68.40 | 76.66 | 88.46 | 90.46 | 34.58 | 45.03 | 18.35 | 29.09 | 52.04 | 61.48 |
| BADGE | 90.58 | 97.16 | **86.44** | _95.10_ | 73.60 | _86.19_ | 48.50 | 63.28 | **89.14** | 95.40 | _69.75_ | 83.02 | 93.18 | 97.67 | 39.13 | 51.15 | 18.58 | 29.03 | 65.60 | 82.70 |
| BASE | 78.69 | 90.60 | 76.05 | 83.42 | 72.26 | 80.02 | 46.21 | 56.96 | 78.18 | 86.86 | 61.26 | 72.95 | 89.43 | 90.66 | 35.86 | 43.07 | 12.82 | 20.97 | 64.32 | 82.06 |
| Confidence | 89.57 | 97.30 | _86.11_ | 94.80 | 76.87 | 85.77 | 46.96 | 63.72 | 86.90 | 93.34 | 69.73 | 83.53 | 93.63 | _97.69_ | 40.95 | 52.54 | 18.23 | 29.71 | _71.96_ | _84.98_ |
| CORESET | _93.10_ | _97.89_ | 85.42 | 94.94 | 77.18 | 85.41 | 44.64 | 56.08 | _87.42_ | _95.44_ | **70.42** | **85.04** | 93.75 | 97.66 | 36.81 | 49.05 | 18.22 | 29.05 | 68.66 | 78.44 |
| Margin | 89.48 | 95.91 | 85.83 | 94.54 | 77.54 | 85.76 | 48.04 | 63.30 | 86.47 | 93.34 | 67.81 | 82.63 | 93.63 | _97.69_ | 38.91 | 50.93 | 18.56 | 28.30 | 69.74 | 80.38 |
| Most Likely | 69.99 | 82.37 | 74.70 | 90.72 | 73.62 | 84.50 | 47.37 | 65.90 | 59.67 | 72.65 | 48.72 | 55.15 | 83.45 | 84.09 | 27.39 | 32.35 | 16.48 | 25.50 | 45.32 | 49.10 |
| GALAXY | 89.19 | 94.27 | 76.99 | 88.23 | 73.41 | 79.17 | 40.15 | 50.18 | 86.87 | 93.81 | 68.74 | 83.23 | 90.03 | 90.44 | 37.17 | 46.94 | 18.40 | 30.59 | 57.51 | 67.50 |
| DIRECT | 64.88 | 67.32 | 71.94 | 82.92 | 69.47 | 78.89 | 42.76 | 57.09 | 70.45 | 76.87 | 59.81 | 67.97 | 90.90 | 94.86 | 38.45 | 49.68 | 18.43 | 29.47 | 67.94 | 78.77 |
| Pretrained Priors | 41.03 | 41.03 | 26.43 | 26.43 | _84.72_ | 84.72 | **67.45** | **67.45** | 43.04 | 43.04 | 6.26 | 6.26 | 52.53 | 52.53 | **91.77** | 91.77 | _64.56_ | _64.56_ | 16.31 | 16.31 |
| **PriorAL (Ours)** | _90.91_ | **97.96** | 82.18 | **95.20** | **86.19** | **86.29** | _67.86_ | **70.31** | 87.12 | **96.20** | 69.71 | _83.93_ | 93.85 | **97.70** | _88.61_ | **93.34** | **66.30** | **74.06** | **73.92** | **85.16** |
| **Balanced Test Accuracy** | | | | | | | | | | | | | | | | | | | | |
| Random | 83.40 | 89.52 | 66.39 | 73.44 | 68.42 | 72.93 | 38.90 | 43.37 | 87.32 | 94.07 | 61.98 | 67.39 | 87.70 | 87.77 | 28.75 | 31.04 | 7.54 | 8.97 | 46.88 | 49.13 |
| BADGE | _93.78_ | **95.96** | **76.42** | _79.15_ | 68.42 | 76.99 | 39.83 | 44.54 | **91.64** | **95.80** | _62.05_ | 68.93 | 88.85 | **91.25** | 30.41 | 32.53 | 7.90 | 9.01 | 51.62 | **59.57** |
| BASE | 78.80 | 93.47 | 67.00 | 73.94 | 68.52 | 72.66 | 39.00 | 43.32 | 81.73 | 89.35 | 57.67 | 65.99 | 89.12 | 89.68 | 29.32 | 29.62 | 6.86 | 7.63 | 54.29 | _58.81_ |
| Confidence | 91.39 | 95.03 | 75.91 | 78.76 | 71.99 | 77.84 | 39.21 | 48.74 | _89.61_ | 93.97 | 62.00 | 69.48 | 89.42 | 89.17 | 30.63 | 32.05 | 7.62 | 7.86 | _56.49_ | 57.26 |
| CORESET | **93.96** | _95.62_ | 75.30 | 78.56 | 72.86 | 77.08 | 37.95 | 46.17 | 89.03 | _95.41_ | **62.11** | 69.81 | **91.05** | 89.58 | 28.36 | 32.57 | 8.18 | 9.09 | 54.31 | 55.64 |
| Margin | 92.23 | 94.99 | _75.99_ | **80.58** | 72.56 | 77.11 | 38.94 | 46.84 | 88.66 | 93.70 | 60.30 | 68.93 | 89.42 | 89.17 | 29.36 | 32.46 | 7.39 | 8.09 | 54.42 | 55.11 |
| Most Likely | 71.09 | 82.98 | 67.17 | 78.09 | 69.93 | 77.56 | 36.97 | 47.96 | 63.37 | 75.94 | 45.74 | 51.48 | 78.74 | 81.77 | 24.54 | 25.76 | 7.22 | 9.17 | 42.30 | 45.24 |
| GALAXY | 92.44 | 94.76 | 67.83 | 77.99 | 69.86 | 73.43 | 33.66 | 40.88 | 89.02 | _95.41_ | 60.91 | _69.87_ | 89.21 | 89.49 | 30.13 | 31.67 | 8.21 | 9.03 | 49.48 | 50.02 |
| DIRECT | 63.35 | 63.71 | 64.21 | 72.74 | 66.33 | 74.68 | 35.06 | 43.98 | 74.56 | 78.43 | 55.95 | 62.63 | 88.25 | _89.79_ | 29.84 | 32.93 | 7.60 | 9.27 | 51.97 | 55.18 |
| Pretrained Priors | 39.82 | 39.82 | 26.60 | 26.60 | _80.21_ | 80.21 | **59.81** | _59.81_ | 47.02 | 47.02 | 6.98 | 6.98 | 53.56 | 53.56 | **43.12** | 43.12 | _14.26_ | _14.26_ | 16.14 | 16.14 |
| **PriorAL (Ours)** | 92.58 | 94.93 | 68.18 | 78.94 | **80.36** | **81.11** | _59.74_ | **61.29** | 88.94 | 94.25 | 61.68 | **70.16** | **89.88** | 89.16 | _40.89_ | _41.52_ | **15.52** | **14.42** | **58.11** | 57.91 |

periments on multiple benchmark datasets. For the text domain, we employ three imbalanced datasets—20NG (Lang, 1995), SST-2 (Socher et al., 2013), and Trec (Li & Roth, 2002)—covering diverse binary/multiclass classification tasks. For the image domain, we adopt imbalanced CIFAR-10 and CIFAR-100 (Krizhevsky et al., 2009). Details on the construction of the imbalanced datasets are deferred to the Appendix B. To examine the robustness of our model in realistic scenarios where label noise naturally occurs, we further conduct experiments on the above datasets with artificially injected label noise. In addition, to examine the feasibility of our method in more challenging, private, and domain-specific scenarios such as medical diagnosis, we also evaluate it on the imbalanced and noisy PathMNIST dataset (Yang et al., 2023), which serves as a representative and challenging benchmark for medical image classification.

**Performance Evaluation.** In this work, we evaluate our method from two aspects: (i)-**Balanced pool accuracy:** Given a pool of training data, we measure the balanced accuracy of our method over the pool to assess its ability to annotate examples under a limited labeling budget, aligning with the goal of transductive learning. (ii)-**Balanced test accuracy**: We measure the balanced accuracy of our method on unseen testing data to assess its generalization performance, reflecting how well the learned model performs beyond the examples in the pool while using a limited labeling budget on oracle annotation. All results are averaged over 4 random runs, with shaded regions in plots indicating 1/2 of a standard deviation.

**Baselines.** We compare our method with two categories of baselines: (1) **Pretrained Priors baseline**: this baseline explicitly uses foundation model, where the small model learns from unlabeled training data with corresponding pseudo-labels generated by the foundation model; we denote it as *Pretrained Priors*. (2) **Classic and State-of-the-Art active learning baselines for imbalanced data**: these baselines are implemented according to their original designs and do not incorporate foundation model information. We compare nine baselines: DIRECT (Nuggehalli et al.), GALAXY (Zhang et al., 2022), BADGE (Ash et al., 2019), BASE (Emam et al., 2021), Confidence Sampling (Settles, 2009), Most Likely Positive (denoted as Most Likely in this work) (Jiang et al., 2018; Warmuth et al., 2001; 2003), Margin Sampling (Scheffer et al., 2001), CORESET (Sener & Savarese, 2017), and Random Sampling.

**Implementation Details.** In the text domain, we adopt both models from Huggingface Transformers (Wolf et al., 2020): the Llama-3.1-8B model as foundation model and the RoBERTa-Base as downstream small model. In the image domain, we employ the CLIP-L14 from open-clip (Cherti et al., 2023) as foundation model and the ResNet-18 model in PyTorch as small model. We refer the readers to Appendix B for more details on our experiment setups.

*Table 2.* Performance comparison under both merge imbalance and long-tailed imbalance settings **with injected label noise**. We report the **balanced pool accuracy and balanced test accuracy** under 10% and 20% annotation cost across text and image domains. Bold numbers denote the best performance for each dataset and second-best results are underlined.

| Method | Merge Imbalance (with Label Noise) | | | | | | | | | | Long-tailed Imbalance (with Label Noise) | | | | | | | | | | | |
| --- | --- | --- | --- | --- | --- | --- | --- | --- | --- | --- | --- | --- | --- | --- | --- | --- | --- | --- | --- | --- | --- | --- |
| | Text Domain | | | | Image Domain | | | | | | Text Domain | | | | | | Image Domain | | | | | |
| | Trec | | AGNews | | C10 | | C100 | | PathMNIST | | Trec | | AGNews | | SST-2 | | C10 | | C100 | | PathMNIST | |
| **Balanced Pool Accuracy** | | | | | | | | | | | | | | | | | | | | | | |
| Random | 50.66 | 59.09 | 45.19 | 52.46 | 48.39 | 55.36 | 29.33 | 36.48 | 61.26 | 61.49 | 55.04 | 63.70 | 49.83 | 58.85 | 64.78 | 68.18 | 28.63 | 35.77 | 16.70 | 27.37 | 38.97 | 45.54 |
| BADGE | 54.10 | 61.73 | 46.76 | 54.90 | 50.16 | 56.87 | 29.37 | 37.14 | 61.99 | 59.52 | 59.27 | 68.04 | 50.85 | 63.02 | 66.19 | 71.14 | 29.63 | 39.37 | 13.42 | 27.81 | **42.38** | 48.86 |
| BASE | 47.49 | 53.40 | 42.50 | 48.85 | 47.11 | 51.05 | 28.01 | 34.47 | 59.77 | 60.43 | 49.56 | 59.04 | 46.74 | 54.87 | 63.20 | 65.36 | 24.10 | 33.71 | 12.27 | 21.02 | 37.53 | 43.04 |
| Confidence | 54.30 | 61.57 | 47.07 | 55.43 | 50.32 | 56.67 | 29.47 | 36.49 | 60.09 | 60.42 | 59.21 | 68.03 | 51.73 | 63.55 | 66.87 | 71.37 | 30.10 | 39.54 | 16.70 | 27.02 | 41.42 | 47.61 |
| CORESET | 54.09 | 62.10 | 47.71 | 55.67 | 50.52 | 56.65 | 29.60 | 37.09 | 62.36 | 62.79 | 58.95 | 69.01 | 51.48 | 63.27 | 66.99 | 71.16 | 29.26 | 38.78 | 16.79 | 27.57 | 41.98 | 50.49 |
| Margin | 54.35 | 61.50 | 46.47 | 55.18 | 49.88 | 55.97 | 29.49 | 36.57 | 60.09 | 60.85 | 57.95 | 67.72 | 52.47 | 63.19 | 66.72 | 71.25 | 28.36 | 38.50 | 16.62 | 27.68 | 37.51 | 48.25 |
| Most Likely | 49.05 | 58.62 | 45.02 | 55.67 | 49.95 | 58.13 | 30.64 | 38.50 | 58.71 | 64.09 | 41.83 | 51.78 | 35.80 | 43.52 | 61.73 | 61.80 | 26.00 | 33.73 | 15.48 | 26.09 | 36.65 | 43.25 |
| GALAXY | 52.10 | 59.52 | 44.90 | 53.08 | 48.56 | 55.10 | 29.39 | 36.49 | 61.21 | 62.99 | 57.53 | 66.41 | 51.79 | 63.15 | 64.75 | 68.24 | 28.33 | 37.56 | 17.31 | 28.45 | 39.04 | 45.51 |
| DIRECT | 47.30 | 54.55 | 43.75 | 51.88 | 48.59 | 54.68 | 29.52 | 36.92 | 59.97 | 56.04 | 46.31 | 56.40 | 43.95 | 50.97 | 64.64 | 69.40 | 28.55 | 38.63 | 16.65 | 24.49 | 42.02 | 50.79 |
| Pretrained Priors | 34.38 | 34.38 | 25.56 | 25.56 | 56.87 | 56.87 | 28.68 | 28.68 | 50.62 | 50.62 | 32.36 | 32.36 | 5.94 | 5.94 | 51.29 | 51.29 | 51.79 | 51.79 | 42.30 | 42.30 | 15.24 | 15.24 |
| **PriorAL (Ours)** | 53.96 | **62.43** | 46.75 | **55.82** | **58.86** | **59.35** | 29.78 | 30.80 | 62.08 | **64.16** | **60.24** | **69.71** | 51.08 | **63.96** | 66.75 | **71.71** | **56.85** | **59.54** | **42.34** | **49.62** | 39.96 | **51.40** |
| **Balanced Test Accuracy** | | | | | | | | | | | | | | | | | | | | | | |
| Random | 69.06 | 75.77 | 59.89 | 60.54 | 54.60 | 58.01 | 28.92 | 32.91 | 74.32 | 77.96 | 74.48 | 79.38 | 54.00 | 59.35 | 80.64 | 80.67 | 23.21 | 23.32 | 6.75 | 7.91 | 35.83 | 38.75 |
| BADGE | 78.86 | 81.60 | 60.72 | 65.92 | 57.36 | 61.60 | 27.21 | 30.29 | 74.10 | 78.16 | 83.28 | 90.12 | 55.67 | 62.45 | 83.51 | **87.68** | 24.18 | 25.76 | 5.22 | 7.17 | 39.45 | 41.56 |
| BASE | 67.62 | 71.98 | 53.73 | 61.13 | 55.32 | 56.50 | 30.03 | 31.77 | 73.91 | 73.55 | 67.65 | 76.10 | 51.50 | 58.46 | 80.42 | 80.73 | 20.99 | 23.93 | 5.86 | 6.83 | 37.48 | 39.42 |
| Confidence | 79.05 | 81.81 | 61.00 | 65.25 | 58.28 | 61.04 | 28.45 | 31.12 | 72.09 | 76.68 | 78.63 | 86.97 | 56.40 | 61.10 | 85.92 | 86.20 | 25.15 | 26.01 | 5.89 | 5.95 | **43.52** | 39.01 |
| CORESET | 72.57 | 84.69 | 62.85 | 68.98 | 58.14 | 59.96 | 29.42 | 32.80 | 76.90 | 81.74 | 80.55 | 86.78 | 55.80 | 60.86 | 84.77 | 83.86 | 23.25 | 25.44 | 5.98 | 6.83 | 41.09 | 41.62 |
| Margin | 79.51 | 82.78 | 60.10 | 68.44 | 56.51 | 58.73 | 28.20 | 31.22 | 72.09 | 80.80 | 76.72 | 84.74 | 56.25 | 62.37 | 84.68 | 85.65 | 22.66 | 24.51 | 6.44 | 6.98 | 35.46 | 39.09 |
| Most Likely | 61.73 | 77.52 | 56.02 | 66.70 | 55.92 | 62.15 | 32.80 | 35.56 | 67.63 | 83.75 | 55.85 | 65.31 | 38.82 | 44.04 | 64.54 | 62.92 | 21.59 | 22.00 | 5.97 | 7.65 | 34.18 | 36.54 |
| GALAXY | 71.83 | 77.30 | 56.59 | 61.92 | 54.16 | 57.74 | 28.62 | 29.58 | 75.37 | **92.03** | 76.97 | 84.87 | 57.76 | 63.43 | 77.40 | 79.46 | 23.46 | 25.02 | 6.44 | 6.52 | 36.09 | 39.30 |
| DIRECT | 56.30 | 60.27 | 54.96 | 62.61 | 53.70 | 56.86 | 27.06 | 29.18 | 69.26 | 61.76 | 59.70 | 69.56 | 50.09 | 53.24 | 80.29 | 82.27 | 23.01 | 25.23 | 6.67 | 7.19 | 39.07 | **44.48** |
| Pretrained Priors | 39.82 | 39.82 | 26.30 | 26.30 | 80.61 | 80.61 | 60.51 | 60.51 | 50.75 | 50.75 | 46.08 | 46.08 | 6.73 | 6.73 | 55.81 | 55.81 | 43.05 | 43.05 | 14.10 | 14.10 | 17.97 | 17.97 |
| **PriorAL (Ours)** | 77.06 | **85.03** | 60.41 | 66.39 | 80.23 | **80.83** | 59.85 | **60.88** | 78.78 | **90.35** | 81.27 | 89.33 | 55.13 | 62.67 | 86.12 | 86.67 | 40.73 | 39.35 | 13.56 | **14.29** | 43.30 | 44.12 |

## 3.2. Main Results

### 3.2.1. EXPERIMENTS UNDER IMBALANCE, WITHOUT LABEL NOISE

Firstly we compare our proposed method against baselines in the image domain. As shown in Table 1, our method outperforms all baselines in terms of balanced pool accuracy and achieves the best or comparable balanced test accuracy. Notably, on the CIFAR-100 dataset with merge imbalance, our method achieves a balanced pool accuracy of 67.86% using only 10% of labeled samples, outperforming all baselines that require 20% labeled samples, thereby leading to a 50% reduction in annotation cost. In addition, under the long-tailed imbalance setting on CIFAR-100 [1], our method surpasses all baselines that rely on 22% labeled samples using only 11% labeled samples. Although our method attains balanced test accuracy close to the top-performing baseline on CIFAR-10 and PathMNIST, it nonetheless delivers the highest balanced pool accuracy after training.

We next evaluate our method in the text domain. From the Table 1, our algorithm generally outperforms the baselines on balanced pool accuracy while maintaining comparable balanced test accuracy. For example, our method yields the best balanced pool accuracy across the two imbalanced settings, except on the long-tailed AGNews dataset, where it achieves the second-best pool accuracy (83.93%) while attaining the highest balanced test accuracy

---

[1] For the long-tailed CIFAR-100 dataset, we record the results under 11% and 22% annotation cost, which are approximately equivalent to 10% and 20% used in the long-tailed CIFAR-10 dataset.

(70.16%). It is worth noting that, in the image domain, foundation models—as reflected by the Pretrained Priors baseline—achieve relatively high prediction accuracy, whereas in the text domain their prediction accuracy is comparatively lower; nevertheless, our method still delivers the best overall performance. For instance, our method achieves the highest balanced pool accuracy of 96.20% on the long-tailed Trec dataset, compared to 95.44% for the second-best CORESET baseline, while Pretrained Priors attains only 43.04%. The results of the Pretrained Priors baseline consume zero annotation cost. Since this baseline does not use any labeling budget, we include its results in the table for comparison; the numbers remain the same across different budget levels. The complete results of active learning process are presented in Appendix C.

### 3.2.2. EXPERIMENTS UNDER IMBALANCE AND LABEL NOISE

We perform a comprehensive set of experiments considering both class imbalance and label noise. In all experiments, we apply a fixed level of label noise by randomly flipping the labels of a given fraction of samples to uniformly selected incorrect classes. As shown in Table 2, we present experimental results on imbalanced datasets with 20% label noise across both image and text domains, except for CIFAR-100 where a 15% label noise rate is used. Based on the results, our method generally surpasses all baselines in balanced pool accuracy while achieving comparable performance on balanced test accuracy. It should be noted that, on the long-tailed CIFAR-100 dataset with injected label noise, our method achieves the best balanced pool accuracy of

42.34% using only 11% of labeled samples, surpassing the performance of all baselines with 22% of data. Notably, it outperforms the best baseline by 17.30% while using 22% of the annotation cost, and achieves the highest balanced test accuracy. We plot the complete active learning process in Figure 3 and we defer the full experimental results to the Appendix C. Based on the results from Table 1 and 2, *our method leverages the rich prior knowledge of the foundation model to deliver consistently superior performance over standard active learning methods. In scenarios where Pretrained Priors is less reliable, the active learning component of our method (Uncertainty-based Sampling in Section 2.3) plays a crucial role in selecting informative samples, thereby mitigating the negative impact of noisy pseudo labels and leading to improved overall performance.*

### 3.3. Analyses and Ablations

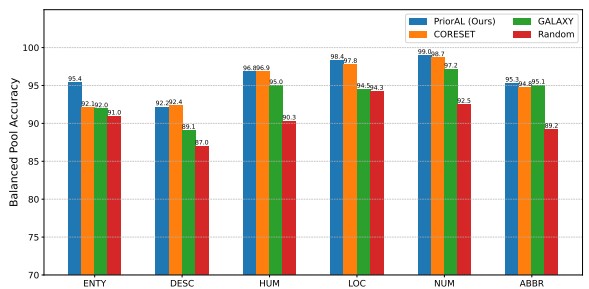

*Figure 2.* Performance of different active learning algorithms across different classes for the Trec dataset without label noise. We plot the balanced pool accuracy under 20% annotation cost. The dataset follows a long-tailed distribution, with class proportions of 38.99%, 24.81%, 17.83%, 8.35%, 6.11%, and 3.92%, respectively.

*Table 3.* Ablation Study on the long-tailed Trec dataset with label noise. We report the balanced pool accuracy and balanced test accuracy under various label-sample budgets with or without IA (Imbalance Awareness). Bold numbers denote the best performance for each metric.

| Labeled Budget | Balanced Pool Accuracy | | Balanced Test Accuracy | |
|---|---|---|---|---|
| | PriorAL | PriorAL (w/o Imbalance Awareness) | PriorAL | PriorAL (w/o Imbalance Awareness) |
| 2% | **34.00** | 34.00 (0.0%) | **43.77** | 43.77 (0.0%) |
| 4% | **46.92** | 46.72 (↓0.4%) | **68.18** | 63.79 (↓6.4%) |
| 6% | **50.99** | 47.77 (↓6.3%) | **72.75** | 64.14 (↓11.8%) |
| 8% | **56.42** | 53.19 (↓5.7%) | **77.00** | 72.96 (↓5.2%) |
| 10% | **60.24** | 56.81 (↓5.7%) | **81.27** | 73.64 (↓9.4%) |
| 12% | **62.76** | 59.57 (↓5.1%) | **82.69** | 78.83 (↓4.7%) |
| 14% | **65.09** | 62.21 (↓4.4%) | **88.41** | 84.16 (↓4.8%) |
| 16% | **65.12** | 63.69 (↓2.2%) | **84.85** | 83.60 (↓1.5%) |
| 18% | **67.92** | 66.20 (↓2.5%) | 87.38 | **87.50** (↑0.1%) |
| 20% | **69.71** | 67.10 (↓3.7%) | **89.33** | 83.49 (↓6.5%) |

#### 3.3.1. EFFECT OF IMBALANCE AWARENESS

To evaluate the contribution of imbalance awareness Eq.(5) in our method, we conduct ablation studies on the Trec dataset. The imbalance awareness aims to assign class-specific weights according to the number of samples in each class. Therefore, we remove the imbalance weight in the equation and perform experiments on the long-tailed Trec dataset with label noise. The results in the Table 3 demonstrate that, without imbalance awareness, the

performance of our method degrades noticeably across the dataset. For example, under a 6% labeled budget, the balanced pool accuracy drops from 50.99% to 47.77%, corresponding to an evident decrease of 6.3%, while the balanced test accuracy decreases by about 11.8%, which highlights the importance and advantage of imbalance awareness under the long-tailed setting.

#### 3.3.2. QUALITATIVE ANALYSIS

To evaluate the effectiveness of our method with respect to class-wise prediction accuracy on an imbalanced dataset, we analyze its behavior across the specific classes on the long-tailed Trec dataset. We conduct the same experiments as in the main experiments, and additionally record the class-wise balanced pool accuracy for our method, CORESET (the best baseline in this setting), GALAXY, and Random, as shown in Figure 2. The dataset is constructed such that class frequencies progressively decrease as the classes are ordered from left to right in the figure. Although the advantage of our method diminishes for minority classes, the results demonstrate that, under the long-tailed class distribution, our method generally achieves the best performance across all classes. For example, in the three minority classes, our method maintains top performance, while the baselines exhibit less consistent or inferior results. Compared to baselines such as GALAXY or DIRECT, which can perform well in the image domain (Zhang et al., 2022), our method consistently achieves superior performance across both image and text domains.

*Table 4.* Performance comparison of our method against baselines on long-tailed CIFAR-10 with asymmetric (class-dependent) noise. We report the recall score on the test set and balanced test accuracy under a 20% annotation budget. Bold and underlined values denote the best and second-best results, respectively.

| Method | Recall Score on the Test Set | Balanced Test Accuracy |
|---|---|---|
| BADGE | 6.87% | 30.78% |
| GALAXY | 6.93% | 30.23% |
| Pretrained Priors | 16.57% | **42.38%** |
| PriorAL (Ours) | **26.47%** | 41.58% |

#### 3.3.3. MOTIVATION OF IMBALANCE AWARENESS

To verify the motivation behind the imbalance awareness Eq.(5), which is based on the intuition that classes with different numbers of samples should contribute unequally—proportionally to their sizes—to the entropy, we conduct additional experiments. The results are shown in Figure 4, where the yellow line represents the results when the contribution is inversely proportional to the number of samples and purple line shows original algorithm without including imbalance awareness. An obvious gap is observed before the labeling budget reaches approximately 2,000. The observation confirms the motivation for implementing imbalance awareness design.

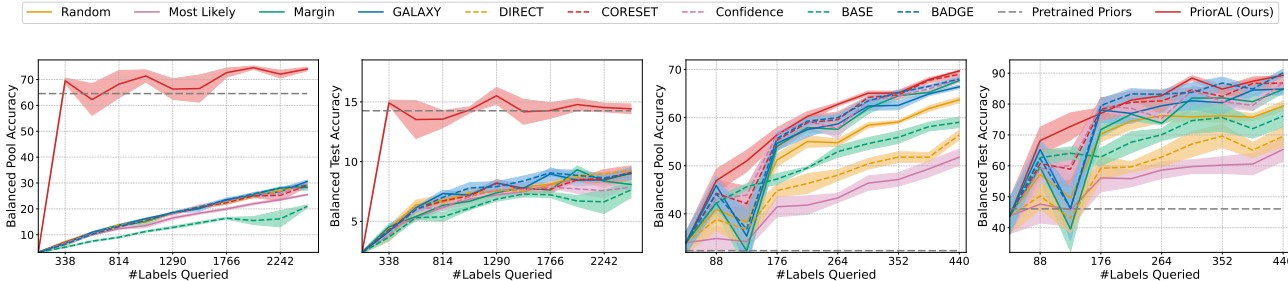

*Figure 3.* Performance comparison of our proposed method against other baselines across image and text domains. The x-axis represents the number of queried labels from the training dataset and y-axis shows the balanced pool/test accuracy. The first two panels show results on the long-tailed CIFAR-100 dataset without label noise. The last two panels present results on the long-tailed Trec dataset with label noise.

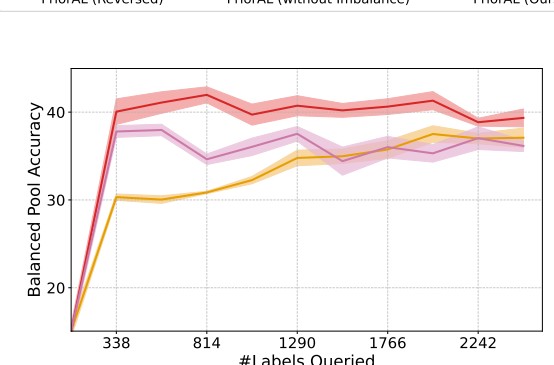

*Figure 4.* Study of our method under different imbalance-aware designs on the long-tailed CIFAR-10 dataset with label noise.

*Table 5.* Performance comparison of our method against baselines in the noiseless but long-tail imbalance setting. The experiments are conducted on the CIFAR-10 dataset using the ViT model. We report balanced pool accuracy and balanced test accuracy under 10% and 20% annotation cost. Bold numbers represent the best performance and second-best results are underlined.

| Method | Balanced Pool Accuracy | | Balanced Test Accuracy | |
|---|---|---|---|---|
| | 10% | 20% | 10% | 20% |
| DIRECT | 33.37 | 48.71 | 26.74 | 29.48 |
| Confidence | 36.15 | 44.67 | 27.25 | 29.03 |
| BADGE | 36.27 | 46.94 | 26.21 | 28.95 |
| Pretrained Priors | 88.98 | 88.98 | **41.81** | **41.81** |
| PriorAL (Ours) | **92.44** | **93.32** | 39.78 | 38.58 |

### 3.3.4. ROBUSTNESS TO ASYMMETRIC NOISE

To evaluate robustness to more complex noise, we further conduct experiments on the long-tailed CIFAR-10 dataset with asymmetric (class-dependent) noise. Following Ghosh & Lan (2021), certain classes are corrupted into semantically similar classes (e.g., BIRD → AIRPLANE), instead of being flipped uniformly at random. We apply 10% noise and report the averaged recall score on the test set and the balanced test accuracy, where the recall score is the macro recall over the three least frequent classes. The results in the Table 4 show that PriorAL achieves the best minority-class recall and remains competitive in balanced test accuracy. These results therefore suggest that PriorAL remains robust when the noise pattern is more challenging than standard random flipping.

### 3.3.5. ROBUSTNESS ACROSS SMALL MODEL VARIANTS

To assess robustness to small model architectures, we conduct additional experiments on the long-tailed CIFAR-10 dataset without label noise, where ViT serves as the small model. We compare our method with several baselines that perform well in our main results; as shown in Table 5. Our method continues to exhibit superior performance in terms of balanced pool accuracy after replacing the small model, while achieving comparable balanced test accuracy in this setting. This robustness can be attributed to the high-quality pseudo labels generated by the foundation model, allowing the performance gains to consistently transfer to different small model architectures.

### 3.3.6. GENERALIZATION ACROSS FOUNDATION MODEL VARIANTS

To evaluate how sensitive our method is to the choice of foundation model, we conduct experiments on the noisy long-tailed CIFAR-10 dataset by replacing the original CLIP-L14 foundation model with SigLIP-B16 and comparing the results. The results in Table 6 show that our method consistently outperforms the classic active learning baseline by a large margin and achieves higher balanced pool accuracy than Pretrained Priors under both foundation models. Our method also achieves comparable balanced test accuracy to Pretrained Priors. As expected, replacing CLIP-L14 with the weaker SigLIP-B16 lowers the performance of both Pretrained Priors and PriorAL, indicating that the benefits provided by the foundation model become weaker. Nevertheless, the qualitative pattern remains stable across the two foundation models, suggesting that our method is not critically dependent on a specific foundation model and instead

degrades gracefully as foundation-model quality decreases.

*Table 6.* Performance comparison of our method with active learning baseline and Pretrained Priors baseline under different foundation models at 20% annotation cost. Bold numbers represent the best performance and second-best results are underlined within each foundation-model pair.

| Method | Balanced Pool Accuracy | Balanced Test Accuracy |
|---|---|---|
| Confidence | 39.54% | 26.01% |
| Pretrained Priors (CLIP-L14) | 51.79% | **43.05%** |
| PriorAL (CLIP-L14) | **59.54%** | 39.35% |
| Pretrained Priors (SigLIP-B16) | 41.59% | **37.02%** |
| PriorAL (SigLIP-B16) | **50.45%** | 34.18% |

### 3.3.7. COMPARISON WITH SEMI-SUPERVISED LEARNING

To further examine whether the gains of PriorAL are mainly attributable to generic semi-supervised pseudo-labeling effects, we additionally compare PriorAL with FlexMatch on the noiseless long-tailed CIFAR-10 dataset. Although FlexMatch is originally designed for conventional semi-supervised learning rather than active learning, we conduct the comparison under a matched protocol: both methods use the same backbone (ResNet-50) and the same number of labeled samples corresponding to a 20% annotation budget, while the remainder of FlexMatch's original semi-supervised pipeline, including its standard pseudo-label generation procedure, is kept unchanged. As shown in Table 7, PriorAL achieves better balanced test accuracy. This result suggests that the improvement of PriorAL cannot be fully explained by a generic pseudo-labeling effect alone, and provides additional evidence that its contribution is distinct from conventional semi-supervised learning methods.

*Table 7.* Comparison between PriorAL and FlexMatch on noiseless long-tailed CIFAR-10. Both methods use ResNet-50 and the same number of labeled samples corresponding to a 20% annotation budget. Bold number denotes the best performance.

| Method | Label Budget | Balanced Test Accuracy |
|---|---|---|
| FlexMatch | 20% | 50.48% |
| PriorAL (Ours) | 20% | **51.49%** |

## 4. Conclusion

In this paper, we present the first study of active learning under both class imbalance and label noise across multiple domains. We propose an efficient active learning algorithm that leverages foundation model priors and imbalance-aware entropy, making it applicable to imbalanced image and text domains with label noise. Experiments on benchmark datasets show that our method reduces annotation requirements by 50% while preserving performance, highlighting the potential of active learning for cost-effective learning under class imbalance and label noise across diverse domains.

## Acknowledgements

Yinglun Zhu acknowledges support from NSF IIS 2425006. Qi Zhang acknowledges support from NSF award 2544949.

## Impact Statement

This paper presents work whose goal is to advance the field of machine learning. There are many potential societal consequences of our work, none which we feel must be specifically highlighted here.

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

# A. Related Work

**Active Learning.**  Active learning aims to build accurate models while reducing annotation cost by iteratively selecting the most informative data for annotation (Settles, 2009). From a theoretical perspective, active learning enjoys provable advantages over passive learning under various settings (Balcan et al., 2006; Hanneke, 2014; Zhang & Chaudhuri, 2014; Krishnamurthy et al., 2019; Zhu & Nowak, 2022a;b). Empirically, it also demonstrate strong performance across a wide range of applications (Tong & Koller, 2001; Kremer et al., 2014; Beluch et al., 2018), especially when combined with deep neural networks (Gal et al., 2017; Sener & Savarese, 2017; Ash et al., 2019; Zhang et al., 2023a). Existing active learning algorithms can be broadly categorized into two classes: uncertainty-based approaches, which prioritize samples with high predictive uncertainty (Settles, 2009; Gal et al., 2017; Ducoffe & Precioso, 2018), and diversity-based approaches (Sener & Savarese, 2017; Geifman & El-Yaniv, 2017; Citovsky et al., 2021), which aim to select representative and diverse subsets of data. Beyond these two paradigms, several works propose hybrid strategies that jointly account for uncertainty and diversity when selecting informative samples (Ash et al., 2019; 2021; Wang et al., 2022; Elenter et al., 2022; Zhang & Zhu, 2025).

**Active Learning with Foundation Models.**  Active learning has become an increasingly prevalent paradigm in the era of foundation models (Yuan et al., 2020; Shelmanov et al., 2021; Gupte et al., 2024), aiming to reduce annotation cost while maintaining strong model performance. Recent studies (Bhatt et al., 2024; Xia et al., 2025) demonstrate that active learning can be effectively combined with large language models (LLMs) to guide data selection and improve LLM training. In the image domain, Gupte et al. (2024) further show that leveraging the rich representations learned by foundation models can substantially enhance active learning performance. More recently, Zhang & Zhu (2025) study multimodal active learning with vision-language models and propose methods to reduce bi-directional annotation costs. Despite these advances, existing work on active learning with foundation models has not systematically examined settings involving class imbalance (Zhang et al., 2022; 2023a) or label noise (Khosla et al., 2022), which are the focus of this paper.

**Active Learning under Imbalance.**  It has become increasingly important in modern applications where data imbalance and rare examples are prevalent. Previous active learning methods, as well as related approaches in active domain adaptation, have mainly focused on selecting annotated data with a more balanced class distribution or improving label-distribution alignment (Aggarwal et al., 2020; Emam et al., 2021; Zhang et al., 2022; Hwang et al., 2022; Nuggehalli et al.). DIRECT (Nuggehalli et al.) offers a modern and effective solution that performs reliably under class-imbalance settings, addressing limitations observed in prior approaches. While DIRECT outperforms prior algorithms, it does not leverage the unlabeled pool for model updates. In addition, its algorithm has been established only in the image domain, despite the growing importance of text-based applications. In contrast, our method leverages scarce labeled data together with abundant unlabeled data to enable more effective model training. Moreover, unlike DIRECT restricted to the image modality, our method can be applied to both image and text domains.

**Active Learning under Label Noise.**  In the field of active learning, label noisy settings remain relatively under-explored. Some related prior works, such as Lin et al. (2016); Younesian et al. (2021), have focused on how to clean existing noisy labels using active learning. Bernhardt et al. (2022) studies active label cleaning in the image domain under a limited relabeling budget, prioritizing samples based on estimated label correctness and sample difficulty. These methods are built on the assumption that all labels provided by the oracle annotator are clean, whereas our work is grounded in the setting where the oracle annotator may provide noisy labels. Other more theoretical active learning studies (Zhang & Chaudhuri, 2015; Chen et al., 2022) primarily aim to detect instances where the weak and strong oracle annotators diverge, enabling selective reliance on the strong annotator for such ambiguous samples. In our setting, we assume the presence of a single annotator, the same assumption made in Nuggehalli et al.. Building on this practically important and widely prevalent setting (Song et al., 2022), our work aims to demonstrate that the proposed method can further improve performance under such challenging but realistic conditions.

# B. Additional Implementation Details

**Extreme Imbalance by Class Merging (Merge Imbalance).**  We construct extremely imbalanced settings for both binary and multi-class classification using popular vision datasets-CIFAR-10, CIFAR-100 (Krizhevsky et al., 2009) and PathMNIST (Yang et al., 2023)-as well as datasets in the text domain including 20NG (Lang, 1995), SST-2 (Socher et al., 2013) and Trec (Li & Roth, 2002). The original forms of these datasets are roughly balanced across 10, 100, or 9 classes in the image domain, and across 20, 2, or 6 classes in the text domain. Except the SST-2 dataset with only two classes, we generate

*Table 8.* Dataset settings for our experiments. The upper block lists datasets constructed via class merging to induce extreme imbalance, while the lower block lists datasets generated under long-tailed distributions. $N$ denotes the total number of training examples in our dataset. $\gamma$ is imbalance ratio defined in the Section 2.1.

| Name | $K$ | $N$ | Imb Ratio $\gamma$ |
|---|---|---|---|
| **Extreme (Merge) Imbalance** | | | |
| Imb CIFAR-10 | 3 | 50,000 | .1250 |
| Imb CIFAR-100 | 5 | 50,000 | .0104 |
| Imb PathMNIST | 2 | 89,996 | .1162 |
| Imb AGNews | 4 | 11,314 | .0497 |
| Imb TREC | 3 | 5,452 | .0209 |
| **Long-Tailed Imbalance** | | | |
| CIFAR-10-LT | 10 | 12,406 | .0100 |
| CIFAR-100-LT | 100 | 10,847 | .0100 |
| PathMNIST-LT | 9 | 21,867 | .0137 |
| AGNews-LT | 20 | 5,187 | .1061 |
| SST-2-LT | 2 | 35,415 | .1892 |
| TREC-LT | 6 | 2,193 | .1006 |

the imbalanced datasets by merging a large number of classes into a single majority class. For example, given an original dataset with $M$ balanced classes, we construct an extremely imbalanced dataset with $K$ classes ($K \ll M$) by retaining the original classes $1, \ldots, K-1$ and merging the remaining classes $K, \ldots, M$ into a single majority class.

**Long-tailed Class Distribution Imbalance (Long-tailed Imbalance).** In addition, we also generate imbalanced datasets under long-tailed distributions. Long-tailed versions are generated for all original datasets across both the image and text domains. Table 8 shows the detailed sizes of the all imbalanced datasets.

**Details on Foundation Model Inference.** In the image domain, the predicted probabilities $p_L$ can be generated in the same manner as in Radford et al. (2021), which provides a link to the relevant code in their paper. In the text domain, $p_L$ can be generated in a zero-shot manner.

**Hyperparameter Settings.** Table 9 lists the hyperparameters used in our experiments across datasets. For each dataset, the same hyperparameters are used for both the label noise and noise-free settings.

*Table 9.* Hyperparameter settings for different datasets.

| Dataset | Epochs | Batch Size | Optimizer | Learning Rate | per-round budget B |
|---|---|---|---|---|---|
| Imb CIFAR-10 | 500 | 100 | Adam | $2 \times 10^{-4}$ | 1000 |
| Imb CIFAR-100 | 500 | 100 | Adam | $2 \times 10^{-4}$ | 1000 |
| Imb PathMNIST | 500 | 100 | Adam | $2 \times 10^{-4}$ | 1799 |
| Imb AGNews | 50 | 128 | Adam | $3 \times 10^{-5}$ | 226 |
| Imb TREC | 50 | 128 | Adam | $3 \times 10^{-5}$ | 109 |
| CIFAR-10-LT | 500 | 100 | Adam | $2 \times 10^{-4}$ | 238 |
| CIFAR-100-LT | 500 | 100 | Adam | $2 \times 10^{-4}$ | 238 |
| PathMNIST-LT | 500 | 100 | Adam | $2 \times 10^{-4}$ | 427 |
| AGNews-LT | 50 | 128 | Adam | $3 \times 10^{-5}$ | 103 |
| SST-2-LT | 50 | 128 | Adam | $3 \times 10^{-5}$ | 708 |
| TREC-LT | 50 | 128 | Adam | $3 \times 10^{-5}$ | 43 |

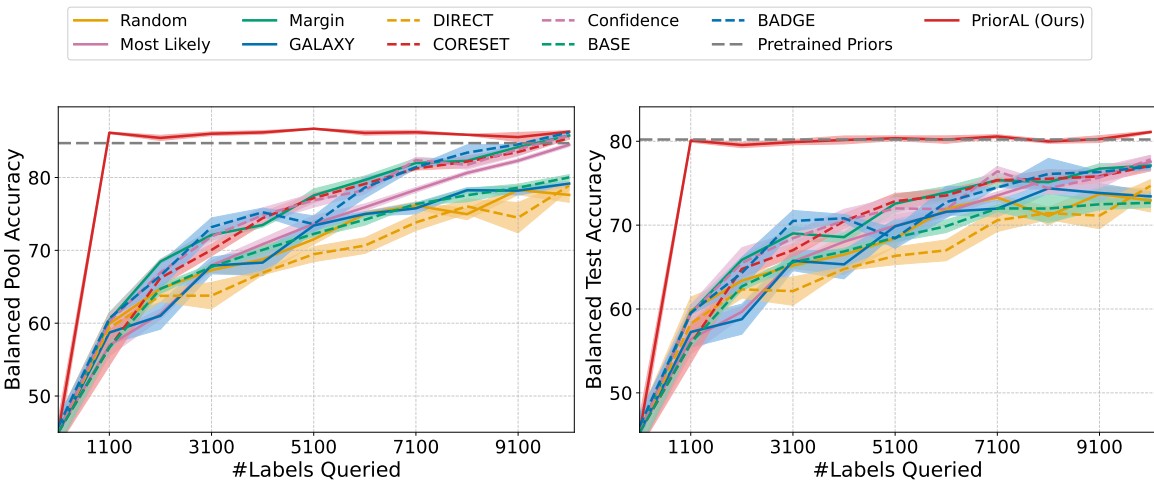

*Figure 5.* Performance comparison of our proposed method against other baselines in the noiseless but merge imbalance setting. The x-axis represents the number of queried labels from the training dataset and y-axis shows the balanced pool/test accuracy. The experiments are conducted on the CIFAR-10 dataset.

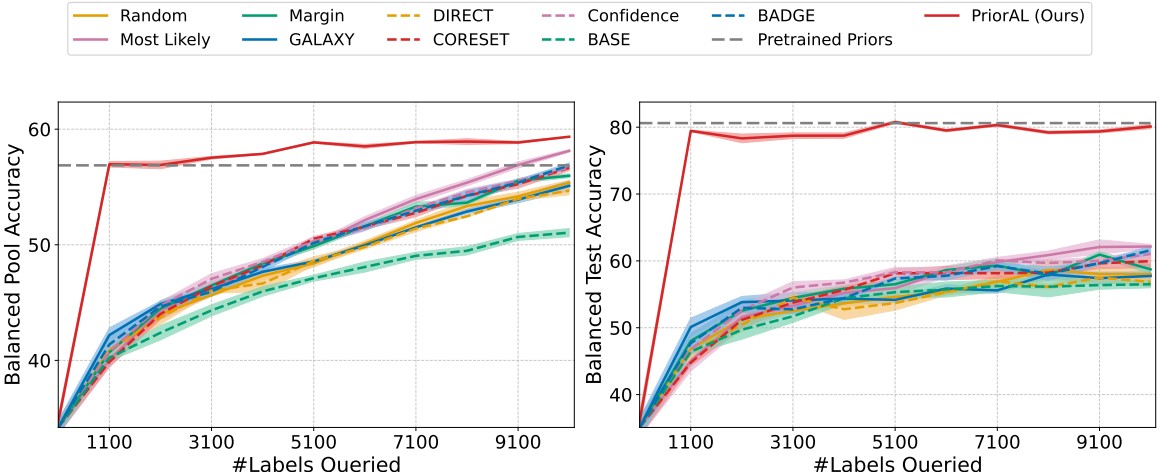

*Figure 6.* Results on the CIFAR-10 dataset with merge imbalance and injected label noise.

## C. All Results

We present all empirical results in Figures 5 to 25. The conclusions from the main results in Section 3 remain consistent across different settings.

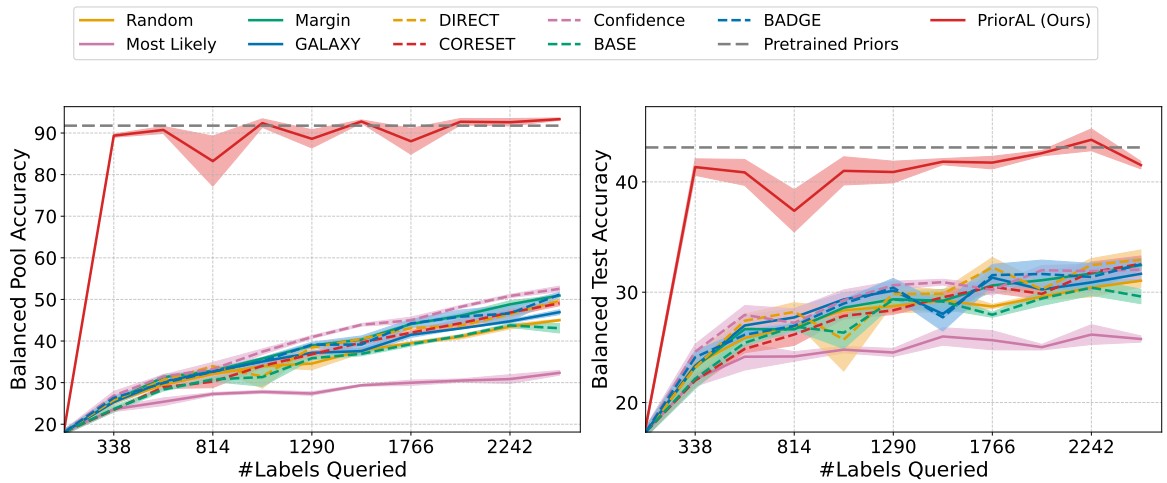

*Figure 7.* Results on the long-tailed CIFAR-10 dataset without label noise.

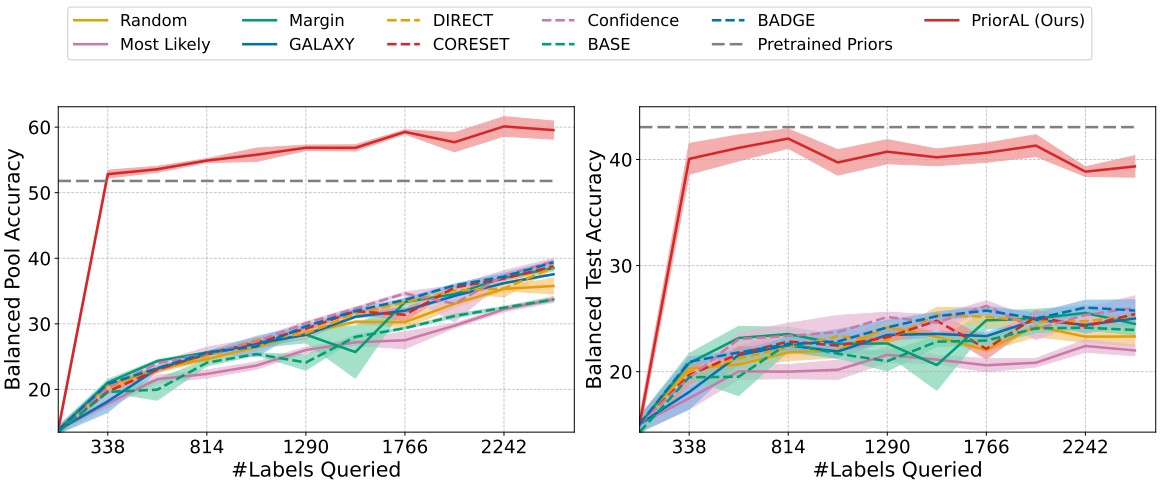

*Figure 8.* Results on the long-tailed CIFAR-10 dataset with injected label noise.

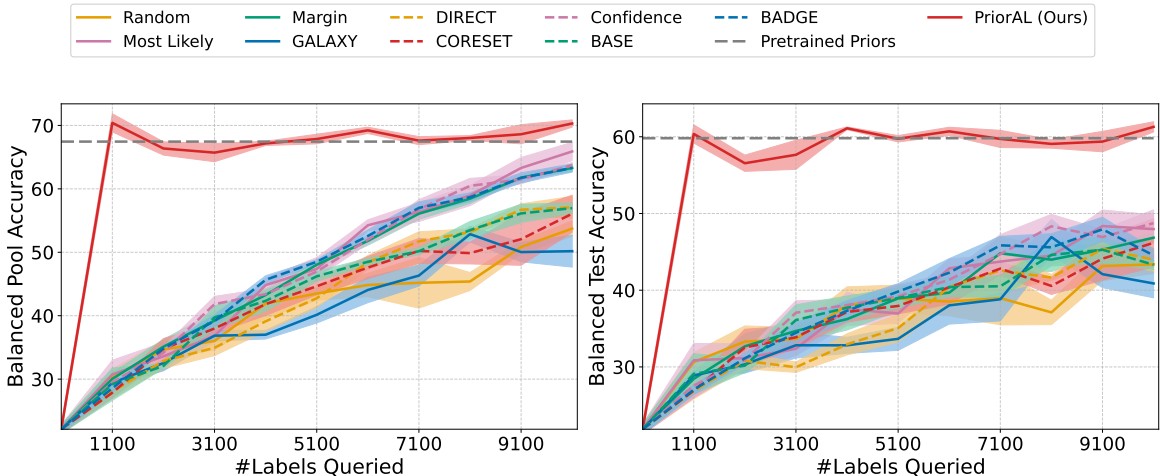

*Figure 9.* Results on the noiseless CIFAR-100 dataset with merge imbalance.

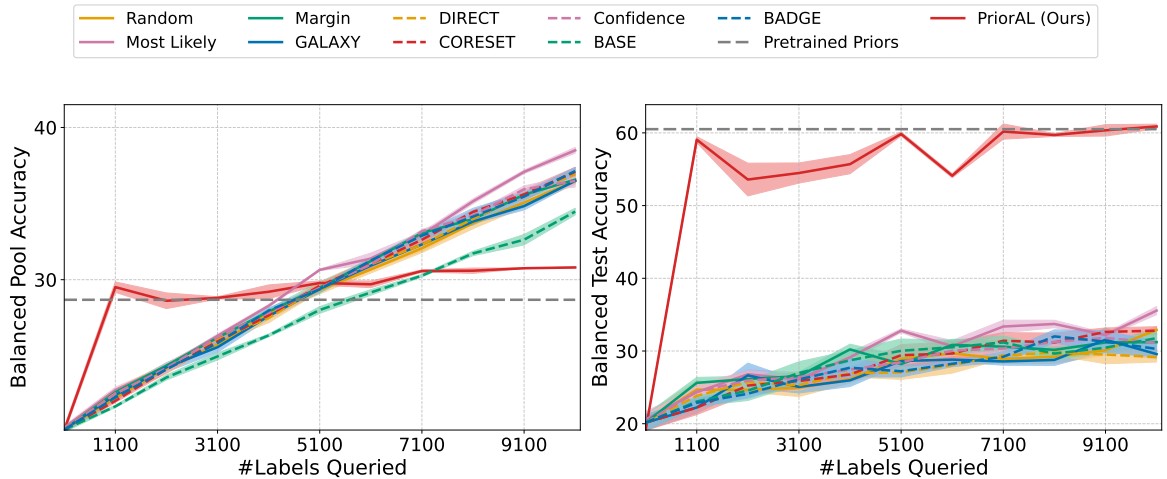

*Figure 10.* Results on the CIFAR-100 dataset with merge imbalance and injected label noise.

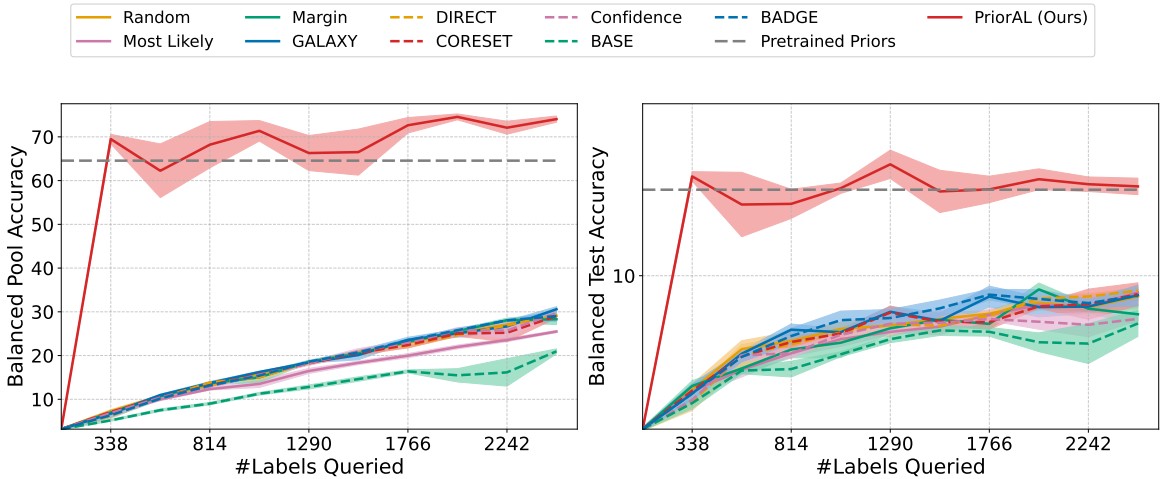

*Figure 11.* Results on the long-tailed CIFAR-100 dataset without injected label noise.

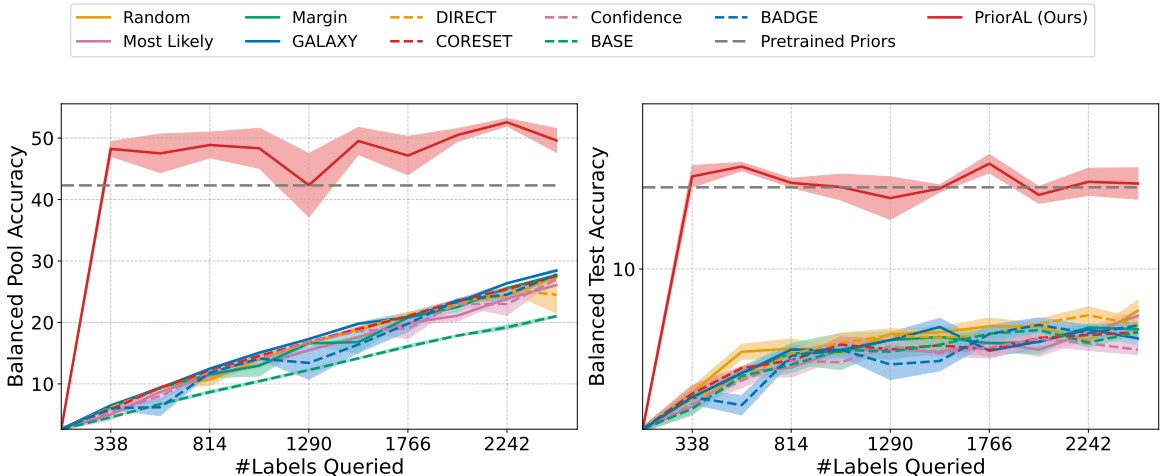

*Figure 12.* Results on the long-tailed CIFAR-100 dataset with injected label noise.

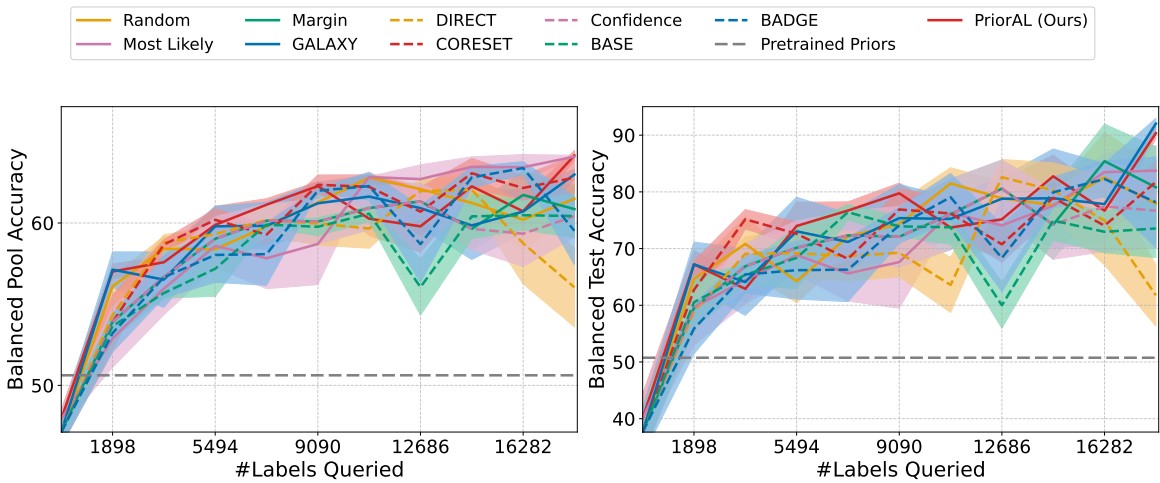

*Figure 13.* Results on the PathMNIST dataset with merge imbalance and injected label noise.

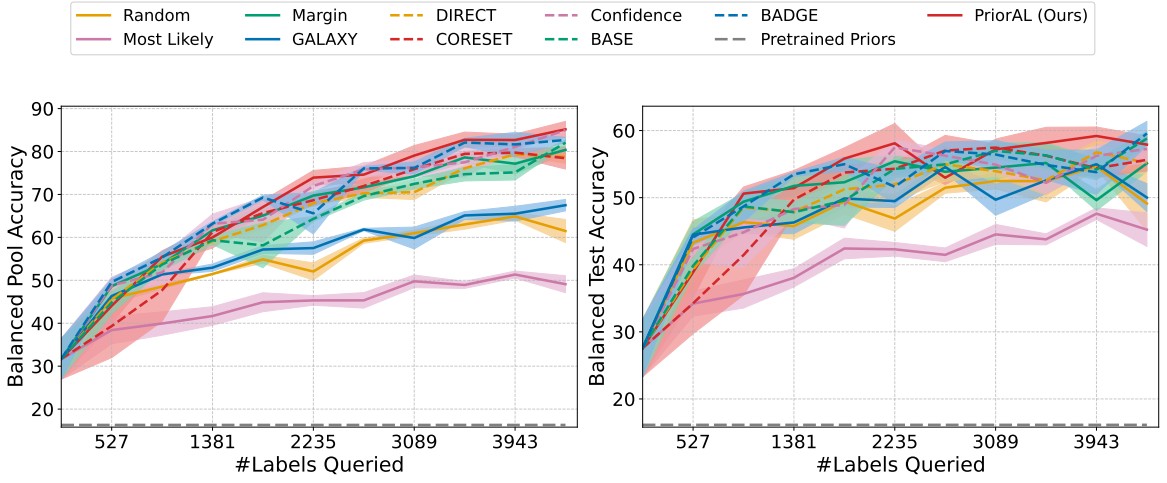

*Figure 14.* Results on the long-tailed PathMNIST dataset without injected label noise.

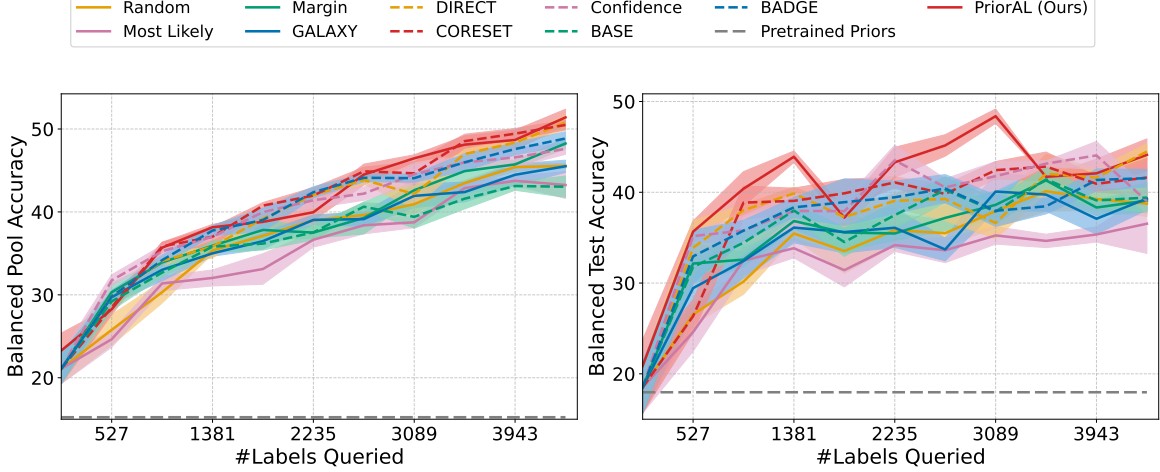

*Figure 15.* Results on the long-tailed PathMNIST dataset with injected label noise.

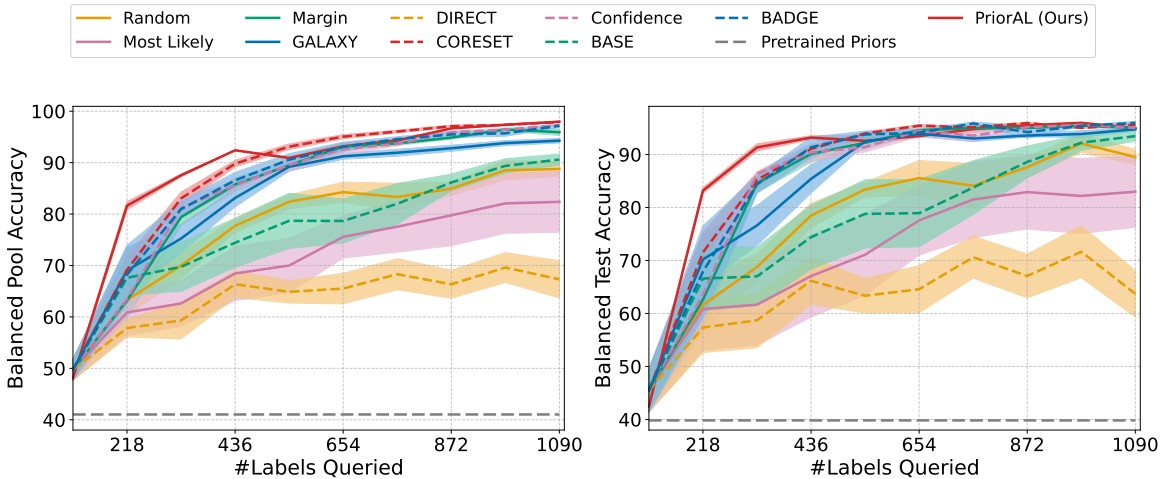

*Figure 16.* Results on the noiseless Trec dataset with merge imbalance.

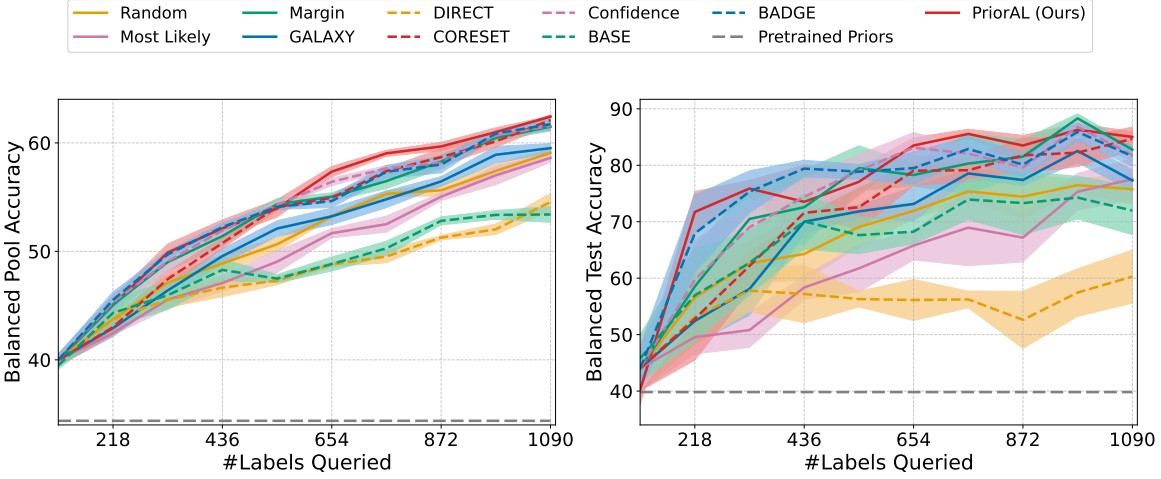

*Figure 17.* Results on the Trec dataset with merge imbalance and injected label noise.

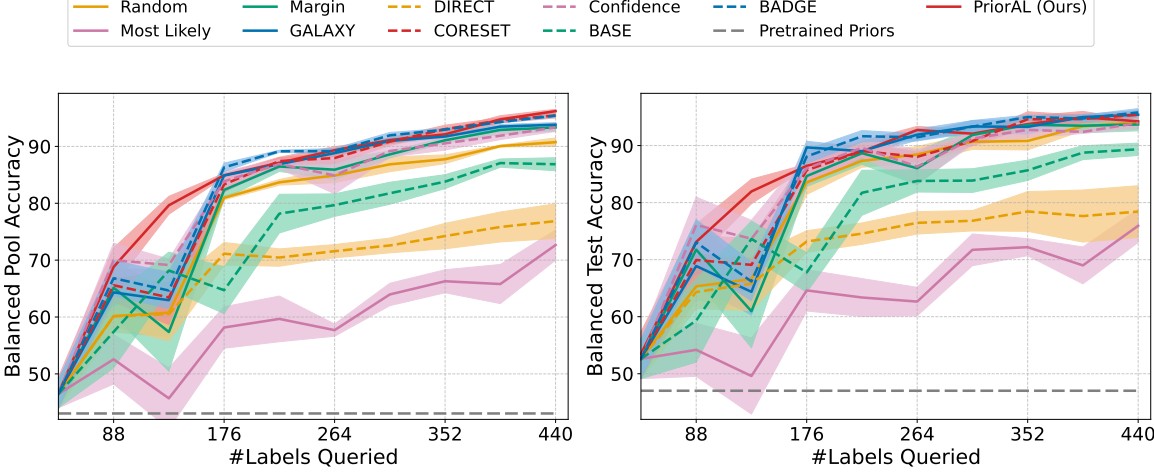

*Figure 18.* Results on the long-tailed Trec dataset without label noise.

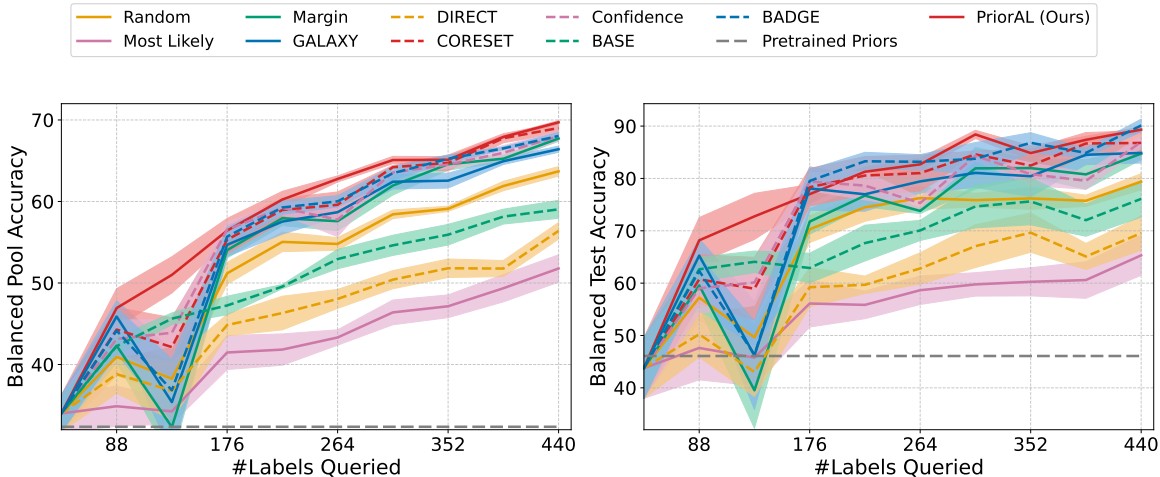

*Figure 19.* Results on the long-tailed Trec dataset with label noise.

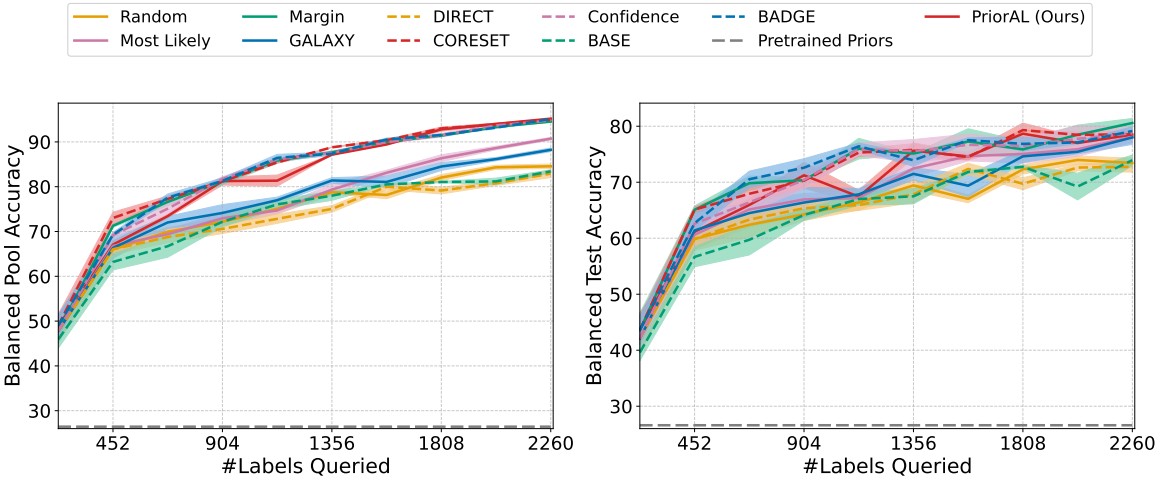

*Figure 20.* Results on the noiseless AGNews dataset with merge imbalance.

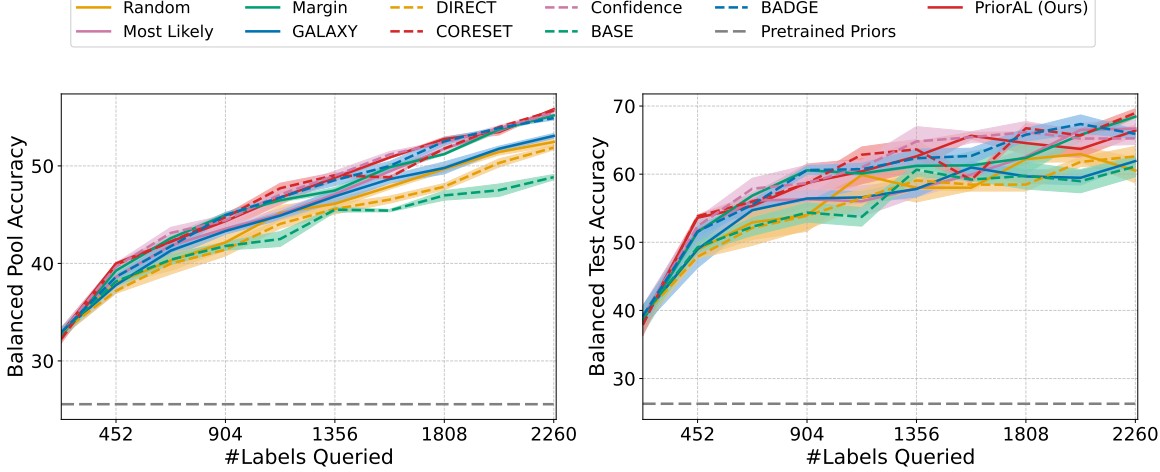

*Figure 21.* Results on the AGNews dataset with merge imbalance and label noise.

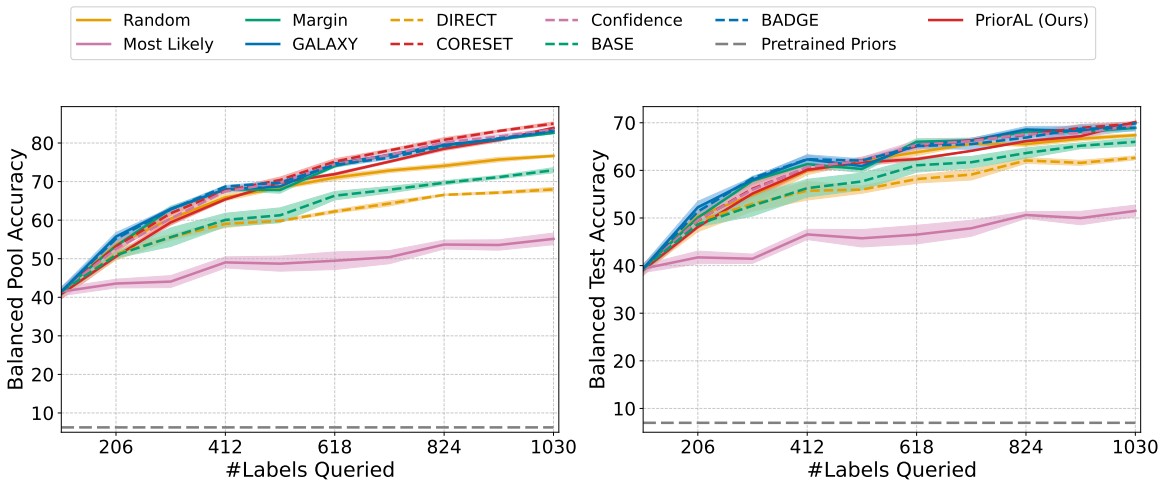

*Figure 22.* Results on the long-tailed AGNews dataset without label noise.

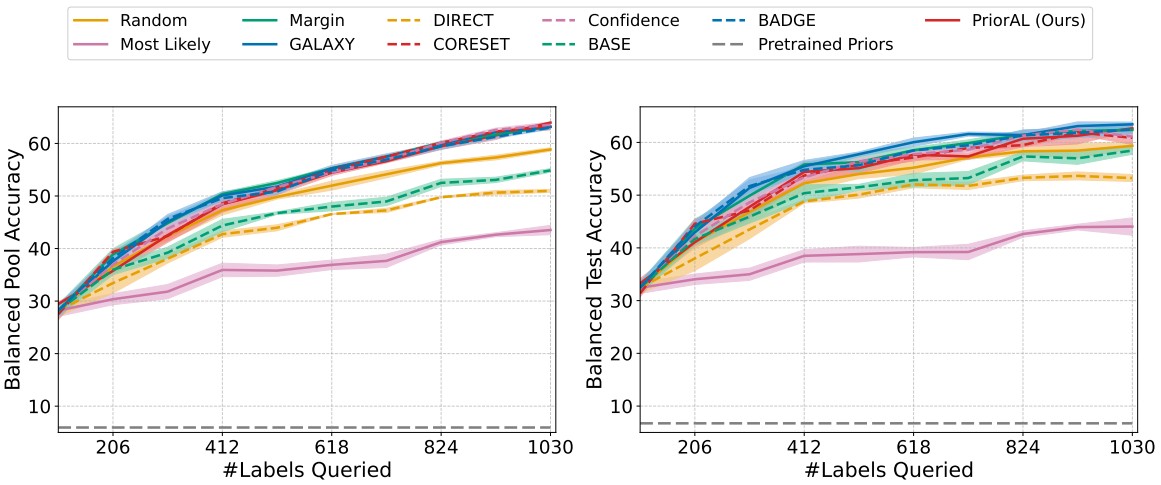

*Figure 23.* Results on the long-tailed AGNews dataset with label noise.

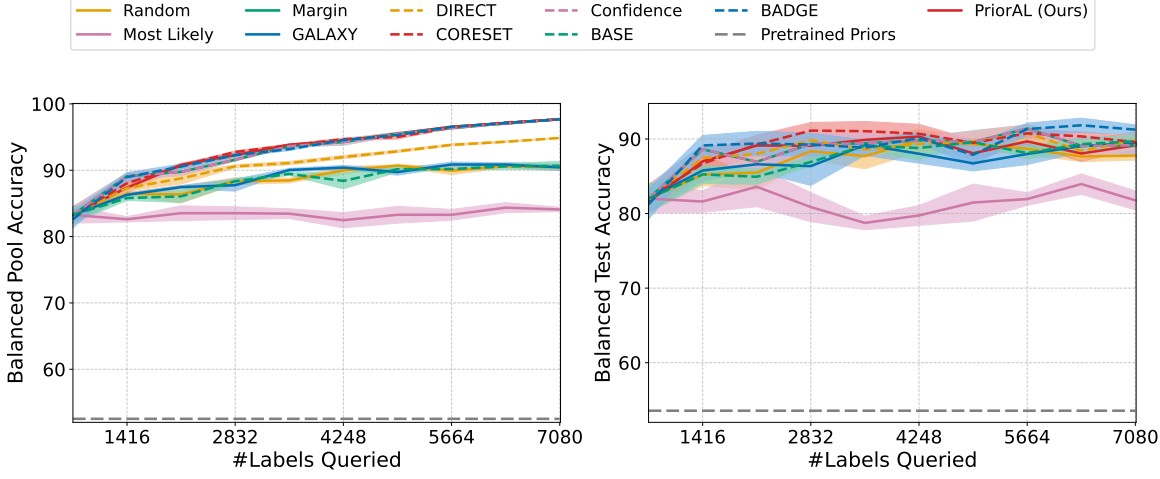

*Figure 24.* Results on the long-tailed SST-2 dataset without label noise.

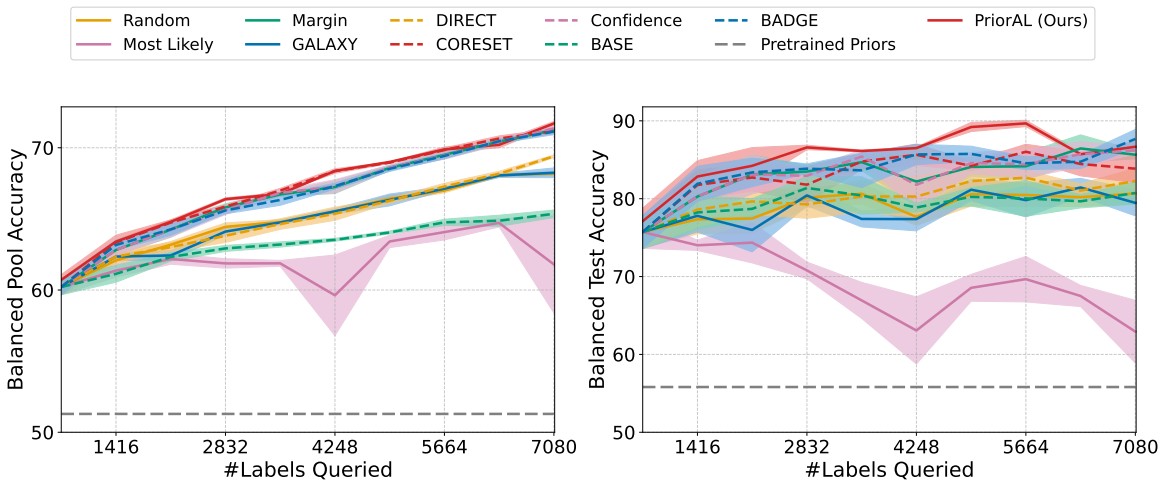

*Figure 25.* Results on the long-tailed SST-2 dataset with label noise.

# D. Other Analyses and Ablations

## D.1. Comparison with the Large Foundation Model

We further compare the foundation model used for prior generation with the trained small model under the same setting. As shown in Table 10, on noisy PathMNIST with merge imbalance, the foundation model achieves 50.06% balanced test accuracy, while the trained small model achieves 90.35%. This result shows that training the small model is not redundant: our framework does not simply inherit the foundation model predictions, but instead exploits foundation model priors to train a more task-adapted model that can match or even exceed the foundation model in challenging noisy and imbalanced settings.

*Table 10.* Comparison between the foundation model and trained small model on noisy PathMNIST with merge imbalance. We report balanced test accuracy, and show balanced pool accuracy in parentheses for reference. Bold numbers indicate the best results.

| Model | Dataset | Balanced Test Accuracy |
|---|---|---|
| Large Model (pool acc: 50.10%) | Noisy PathMNIST with Merge Imbalance | 50.06% |
| Small Model (pool acc: **64.16%**) | Noisy PathMNIST with Merge Imbalance | **90.35%** |

*Table 11.* Sensitivity analysis of the hyperparameter $\rho$ on noiseless Trec dataset in terms of balanced pool accuracy and balanced test accuracy at 20% annotation cost. Bold numbers indicate the best results.

| $\rho$ | Balanced Pool Accuracy | Balanced Test Accuracy |
|---|---|---|
| 99% | **96.20%** | **94.25%** |
| 90% | 82.07% | 79.45% |
| 80% | 77.35% | 76.53% |

## D.2. The Impact of $\rho$

When the foundation model prior is accurate for the underlying datasets, as in the majority of the cases, we adopt a small-to-moderate value of $\rho$ to better utilize pseudo-labeled samples. In cases where the foundation model priors are not accurate for the underlying dataset, we use a larger $\rho$ to allow PriorAL to learn better from the data annotation oracle. Table 11 provides an ablation study for the later case where foundation model priors tend to be inaccurate for the underlying dataset (the long-tailed Trec dataset without label noise), where higher values of $\rho$ lead to better results.

## D.3. Empirical Runtime Comparison

To assess the practical data selection time of PriorAL, defined as the time spent determining how to select unlabeled examples for annotation, we compare it with BADGE on the noiseless Trec dataset with merge imbalance under the same setting. We report their time across 20 active learning iterations and average the results over four random seeds. For PriorAL, the large-model prior is computed only once, while the remaining selection pipeline is executed over 20 iterations. PriorAL requires less data selection time (221.40s) than BADGE (176.68s), indicating that its selection procedure is relatively efficient.

