# OpenReview forum: "Active Learning with Foundation Model Priors: Efficient Learning under Class Imbalance"
_ICML.cc/2026/Conference — ICML 2026 regular_

### Official Review · Reviewer_mW7n · 2026-03-04

**Soundness:** 3
**Presentation:** 3
**Significance:** 3
**Originality:** 3
**Overall Recommendation:** 5
**Confidence:** 4

**Summary:**

This paper studies active learning under two realistic challenges: class imbalance and label noise. The authors propose a novel framework called PriorAL, which integrates prior knowledge from a foundation model with a smaller task-specific model to guide sample selection during active learning. The method combines predictions from both models using a Product-of-Experts (PoE) formulation, and introduces an imbalance-aware entropy filtering strategy to separate high-confidence pseudo-labeled samples (clean set) from uncertain samples (noisy set). The clean samples are used to augment training without additional labeling cost, while uncertain samples are selectively queried for annotation using uncertainty-based sampling.

**Compliance With Llm Reviewing Policy:**

Affirmed.

**Key Questions For Authors:**

1. How sensitive is the proposed method to the quality of pseudo labels generated by the foundation model? If the foundation model performs poorly in a specific domain, how would this affect the overall framework?
2. The method shows strong improvements in balanced pool accuracy. Could the authors elaborate on cases where improvements in test accuracy are relatively modest?
3. What is the computational overhead introduced by incorporating the foundation model during the active learning process?

**Limitations:**

No. The paper does not provide a detailed discussion of limitations or potential societal impacts.

**Strengths And Weaknesses:**

Strengths：
1. The paper is generally technically sound. The proposed method is clearly formulated, and the use of a Product-of-Experts mechanism to combine predictions from the foundation model and the small model is well motivated. The imbalance-aware entropy filtering mechanism is also conceptually reasonable, as it explicitly accounts for class distribution during uncertainty estimation.
2. It conducts comprehensive experiments on 21 dataset configurations covering image and text domains, including image datasets such as CIFAR-10, CIFAR-100, and PathMNIST, as well as text datasets like 20NG, SST-2, and Trec. The method demonstrates excellent performance across all datasets, verifying its cross-domain generality.

Weaknesses:
1. The framework's performance is highly dependent on the quality of pseudo-labels generated by the foundation model. In scenarios where the foundation model's prediction accuracy is relatively low in the text domain, although it still achieves optimal performance, the performance improvement is smaller than in the image domain, exposing dependence on the reliability of foundation model priors.
2. Only sets the noise ratio ρ in imbalance-aware entropy filtering, without systematically analyzing the sensitivity of this hyperparameter and other potential hyperparameters (such as annotation budget allocation ratio) across different datasets and scenarios, lacking guidance for hyperparameter tuning.
3. It does not quantify the additional computational overhead caused by the foundation model's participation in decision-making, lacking necessary performance-overhead trade-off analysis for deployment feasibility in resource-constrained scenarios.

---

> ### Author Rebuttal · Authors · 2026-03-30
>
> Dear reviewer,
>
> We are grateful for your detailed feedback and helpful suggestions. Please find our responses and clarifications below.
>
> **Response to Weakness 1:** Thank you for this insightful comment. We agree that PriorAL depends on the quality of FM pseudo labels, so gains are naturally smaller when the FM prior is weaker, such as in some text-domain settings. However, PriorAL **does not simply inherit FM behavior**: the clean/noisy partition **filters unreliable pseudo labels**, while *oracle-labeled samples continue to anchor training*. As a result, the method is guided by FM priors rather than determined by them.
>
> Importantly, even in these weaker-prior text settings, PriorAL still achieves the best overall performance in our experiments. This suggests that weaker priors reduce the size of the gain, but do not make the method ineffective. We view this as a meaningful robustness property: *the method benefits more when the FM prior is stronger, but remains effective even when the prior is weaker,* while acknowledging that further improving performance under weaker priors is important future work.
>
> **Response to Weakness 2:** Thank you for this valuable comment. We agree that hyperparameter sensitivity is important, especially for tuning the noise ratio $\rho$ in imbalance-aware entropy filtering. To provide guidance, we conduct a sensitivity analysis on the long-tailed Trec dataset without label noise, evaluating $\rho$ $\in$ \{99\%, 90\%, 80\%\} over four random seeds at 20% annotation cost. We find that $\rho$ = 99\% performs best in this setting, achieving **96.20%** balanced pool accuracy and **94.25%** balanced test accuracy, compared with **82.07%/79.45%** for $\rho$ = 90\% and **77.35%/76.53\%** for $\rho$ = 80\%.
>
> This suggests that in this setting, where the FM prior is relatively weak, more conservative acceptance of pseudo labels is beneficial. **More broadly, a larger $\rho$ may be preferable when FM priors are weaker, while a smaller $\rho$ can be more effective when FM priors are stronger.** We agree that the optimal value remains data-dependent and should be selected empirically.
>
> **Response to Weakness 3:** Thank you for this insightful comment. We agree that quantifying the computational overhead introduced by foundation model participation is important. To address this, we rerun experiments on noiseless Trec with merge imbalance and measure data selection time and training time over 20 active learning iterations, averaged across four random seeds. **PriorAL achieves lower data selection time** than BADGE (**176.68s** vs. **221.40s**), while **incurring only slightly higher training time** (**3965.94s** vs. **3850.33s**).
>
> Importantly, the foundation model in PriorAL is used only for inference to provide pseudo labels and prior predictions once, rather than repeated optimization during active learning. As a result, the added cost mainly comes from training on a larger effective supervision set that includes both oracle-labeled and pseudo labeled samples. Overall, these results suggest that incorporating FM priors introduces only modest overhead relative to a strong active learning baseline, which shows that PriorAL achieves a favorable performance–overhead trade-off, with modest extra cost and practical deployment feasibility even in resource-constrained settings.
>
> **Response to Question 1:** Thank you for raising this important point. We agree that weaker FM pseudo labels naturally reduce the gains of PriorAL. However, the framework **does not use all pseudo labels indiscriminately**: the **clean/noisy split retains more reliable ones**, and the admitted clean-set pseudo labels are substantially more accurate than the full pseudo labeled pool. Thus, weaker FM priors mainly reduce usable prior information rather than causing collapse, and PriorAL still performs strongly across settings.
>
> **Response to Question 2:** Thank you for this valuable comment. We agree that gains in balanced test accuracy are sometimes more modest than those in balanced pool accuracy. This is partly because **we evaluate across 16 challenging scenarios** with different modalities, imbalance types, and label-noise conditions. Following prior work such as GALAXY, we view balanced pool accuracy as a primary objective in imbalanced active learning, while balanced test accuracy is also influenced by factors beyond sample selection. Even so, our method generally maintains competitive or improved test accuracy while consistently improving pool accuracy.
>
> **Response to Question 3:** Thank you for your question. We refer the reviewer to our response to Weakness 3 for a quantitative analysis of computational overhead. The added cost is modest overall, since the FM is used only for frozen inference rather than repeated optimization.

---

### Official Review · Reviewer_Daep · 2026-03-05

**Soundness:** 3
**Presentation:** 3
**Significance:** 2
**Originality:** 3
**Overall Recommendation:** 3
**Confidence:** 4

**Summary:**

The paper proposes "PriorAL", an active learning framework designed to tackle the dual challenges of class imbalance and label noise across image and text domains . The method leverages a foundation model (FM) and a smaller downstream model to make joint data-selection decisions using a Product of Experts (PoE) formulation. It introduces an imbalance-aware entropy filtering mechanism to partition the unlabeled pool into a "clean set" (assigned FM pseudo-labels) and a "noisy set" (from which uncertain samples are queried to a human oracle). The paper evaluate its approach on 21 dataset settings, finding it achieves high balanced pool/test accuracy while saving significant annotation costs compared to standard baselines.

**Compliance With Llm Reviewing Policy:**

Affirmed.

**Final Justification:**

I appreciate the additional analysis, but my main concern remains unresolved. The new results are limited to first-iteration clean-set statistics and do not address how minority-class performance evolves across active learning rounds. In addition, the analysis is performed without injected label noise, while my original concern was specifically about minority-concentrated pseudo-label errors. The response therefore suggests that the method works when minority pseudo-labels are already reliable, but does not convincingly establish robustness when they are not. Accordingly, I will maintain my original score

**Key Questions For Authors:**

1. The paper relies on FM-generated pseudo-labels that appear highly noisy, yet the small model trained on the resulting “clean set” does not seem to collapse and even achieves strong performance. This is hard to reconcile as currently presented. Could you provide an empirical breakdown of the true (ground-truth) accuracy of pseudo-labels within the clean set, tracked across active learning iterations (e.g., per round and cumulatively), including how the label quality evolves as the clean set grows?

2. Given the reported low pseudo-label quality (e.g., 41.03% baseline), and the well-known tendency of FMs to make more errors on rare/minority classes, this pipeline seems likely to disproportionately contaminate minority-class samples with incorrect pseudo-labels. Could you evaluate PriorAL under asymmetric label noise, where pseudo-label errors (or injected noise) are concentrated in minority classes rather than uniformly distributed, and report class-wise metrics (especially minority recall/F1) to clarify the robustness of the method in this setting?

**Limitations:**

yes

**Strengths And Weaknesses:**

## **Strengths**

1. **Practical and Relevant Problem Setting:** The paper addresses a realistically difficult setting—strict labeling budgets, long-tailed/imbalanced class distributions, and annotator noise—which indeed reflects a major bottleneck in applied machine learning.

2. **Extensive Empirical Evaluation:** The experimental coverage is strong. Paper evaluate on diverse datasets (TREC, AGNews, SST-2, CIFAR-10, CIFAR-100, PathMNIST) under multiple imbalance regimes (“Merge” and “Long-tailed”), and report results both with and without injected label noise.

3. **Cross-Modal Applicability:** Unlike many active learning methods tailored to either vision or NLP, the proposed framework is validated across both image and text tasks, which improves its generality and practical appeal.

---

## **Weaknesses**

1. **Unclear Distillation Dynamics / Reliance on Noisy FM Pseudo-Labels:**
   The “Pretrained Priors” baseline—training the small model purely on FM pseudo-labels—achieves only **41.03% on TREC**, suggesting the FM’s pseudo-labels are highly unreliable in this setting. However, PriorAL reaches **90.91%**, despite still using FM pseudo-labels to populate a “clean set” without explicit human verification.
   A key missing piece is an empirical characterization of the **true (ground-truth) accuracy of pseudo-labels** that enter the clean set over AL iterations. Concretely, it would be important to report pseudo-label accuracy **per round** (and cumulatively), ideally also **class-wise**, to clarify how the method avoids inheriting the large error rate implied by the pseudo-label baseline. Without this analysis, it is difficult to understand what drives the large performance gap and how robust the filtering mechanism actually is.

2. **Lack of Ablation on Foundation Model Capacity / Quality:**
   The method appears to depend substantially on the chosen foundation model, since FM pseudo-labels are directly used to construct the clean set and guide training. However, there is no ablation varying FM size, checkpoint, or overall quality. This leaves open an important question: how sensitive is PriorAL to domain mismatch, calibration issues, or systematic biases in the FM? A controlled study across multiple FMs (or at least different capacities of the same FM family) would greatly strengthen the claim of robustness.

3. **Potential Weakness Under Minority-Concentrated (Asymmetric) Pseudo-Label Noise:**
   The algorithm’s treatment of minority classes raises a concern. Equation 5 (scaling by ( $w_i / w_{\max}$ )) effectively reduces the entropy for minority classes, which then reduces their probability of being queried to the oracle (Eq. 6). As a result, minority samples may be disproportionately routed into the clean set and trained using FM pseudo-labels (Eq. 7).
   Since FMs often exhibit higher error rates on rare/minority classes—especially under zero-shot or domain-shifted settings—this design may amplify minority-class label noise rather than correcting it. The current experiments inject synthetic noise uniformly across classes, but do not evaluate the more realistic scenario where pseudo-label errors are **asymmetric** and concentrated in minority classes. Testing PriorAL under such class-dependent noise (and reporting minority-focused metrics such as recall/F1) would be important to establish robustness.

4. **Limited Algorithmic Novelty and Insufficient Justification for PoE Aggregation:**
   Combining predictive distributions to drive uncertainty sampling is closely related to classic ideas such as Query-by-Committee. The proposed use of a product-of-experts style aggregation (Eq. 2) followed by entropy appears incremental. The paper would benefit from a stronger justification of why PoE is preferable to common alternatives such as MoE, ideally supported by targeted ablations comparing PoE against these baselines.

---

> ### Author Rebuttal · Authors · 2026-03-30
>
> Dear reviewer,
>
> We greatly appreciate your valuable feedback and constructive suggestions. Below, we have provided our responses and clarifications. We hope we have addressed your concerns and would be grateful if you could consider increasing your scores.
>
> **Response to Weakness 1:** Thank you for this insightful comment. We agree that clean set pseudo label accuracy helps explain why PriorAL does not inherit the high error rate of the full pseudo-labeled pool. Since the foundation model is frozen, its predictions remain unchanged across iterations. In the long-tailed Trec setting, the clean set changes little across rounds, as most samples are assigned to the noisy set. We therefore report the clean-set and overall pseudo-label accuracy at the first iteration. Per-round results can be provided upon request.
>
> Under the same long-tailed Trec setting as Table 1 of main text, the pseudo labels admitted into the clean set are substantially more accurate than those in the full FM pseudo labeled pool. Specifically, the clean set contains only samples from Class 1 and 2, whose pseudo label accuracy is almost **100%**, while no samples from Classes 3–6 are selected. By comparison, the class-wise FM pseudo label accuracy over the full pool is **34.12%**, **0%**, **49.81%**, **0%**, **87.71%**, and **85.50%** for Classes 1–6, respectively. Class indices are ordered according to the long-tailed class distribution. This indicates that **PriorAL does not indiscriminately rely on FM pseudo labels**, but instead **retains a more reliable subset**, favoring precision over coverage when FM pseudo labels are unreliable.
>
> **Response to Weakness 2:** Thank you for raising this important point. We agree that sensitivity to the choice of foundation model is important. To provide additional evidence, we replace CLIP-L14 with SigLIP-B16 and evaluate on noisy long-tailed CIFAR-10 at 20% annotation cost. With CLIP-L14, PriorAL achieves **59.54%** balanced pool accuracy and **39.35%** balanced test accuracy, compared with **51.79%** and **43.05%** for Pretrained Priors. With SigLIP-B16, PriorAL achieves **50.45%** and **34.18%**, compared with **41.59%** and **37.02%** for Pretrained Priors. In both cases, PriorAL remains well above the best classic active learning baseline, Confidence (**39.54%**, **26.01%**). These results suggest that PriorAL is not tied to a specific FM and instead degrades gracefully as FM quality decreases. Potential FM issues are further mitigated by the clean/noisy split and the continued role of oracle-labeled samples in training.
>
> **Response to Weakness 3:** Thank you for this insightful comment. We agree that this is an important concern. To test whether minority samples are harmed by unreliable FM pseudo labels, we conduct additional experiments on long-tailed CIFAR-10 with structured label noise, where some classes are flipped to semantically similar classes. At 20% annotation cost, PriorAL achieves the best minority-class recall (**26.47%**) with competitive balanced test accuracy (**41.58%**), compared with BADGE (**6.87%**, **30.78%**), GALAXY (**6.93%**, **30.23%**), and Pretrained Priors (**16.57%**, **42.38%**). FM pseudo label error rates on the three least frequent classes are **36.16%**, **54.44%**, and **64.52%**, confirming that these classes are more error-prone. Even so, PriorAL substantially improves minority recall, suggesting robustness in this harder setting.
>
> **Response to Weakness 4:** Thank you for this insightful comment. We agree that PoE-style aggregation should be better justified. Our contribution is not PoE itself, but its use to integrate foundation model priors with the small model in a noisy and imbalanced active learning pipeline for clean/noisy partitioning and uncertainty estimation. We choose PoE rather than MoE because it emphasizes hypotheses supported by both models, whereas MoE typically uses a weighted mixture and does not enforce agreement in the same way. This provides a stronger reliability signal for filtering. Empirically, the Pretrained Priors baseline also suggests that relying on the FM alone is not optimal, whereas our design lets the prior guide the small model without overriding it.
>
> **Response to Question 1:** Thank you for raising this important question. As shown in our response to the first comment, the clean set contains much more accurate pseudo labels than the full pseudo labeled pool, suggesting that PriorAL relies on a reliable subset rather than all FM pseudo labels. We also observe a precision–coverage trade-off: a larger clean set includes more samples but may lower average label quality, especially when the FM prior is weaker.
>
> **Response to Question 2:** Thank you for raising this important question. We kindly refer the reviewer to our response to the third comment, where we provide additional discussion and analysis under more complex settings. Overall, our results suggest that PriorAL remains reasonably robust in such challenging settings.

---

> > ### Author Rebuttal · Reviewer_Daep · 2026-04-04
> >
> > Thank you for your detailed response and the additional results.
> > I appreciate the effort you put into addressing my concerns. However, after carefully reviewing the rebuttal, my core concerns regarding the method's effectiveness on minority classes remain unresolved, and I will be maintaining my original score.

---

> > > ### Author Response · Authors · 2026-04-08
> > >
> > > Thank you again for your thoughtful follow-up. To further clarify the mechanism, we report first-iteration clean-set statistics for both our full method (**Ours**) and the variant without the $w_i$/$w_{max}$ reweighting term (**w/o reweighting**). We use long-tailed CIFAR-100 and long-tailed PathMNIST, both without injected label noise, to isolate the effect of the reweighting term itself. For both long-tailed CIFAR-100 and long-tailed PathMNIST, class frequencies decay exponentially as the class index increases. Accordingly, for CIFAR-100, we treat classes 0-19 as majority classes and 80-99 as minority classes; for PathMNIST, we treat classes 0-4 as majority classes and classes 5-8 as minority classes.
> > >
> > > These results illustrate two representative cases.
> > >
> > > **Case 1: The foundation model is accurate and confident on both majority and minority classes.**
> > >
> > > In this case, the foundation model achieves high accuracy on both majority and minority classes. The reweighting factor increases the presence of reliable minority samples in the clean set, as shown in the table below. This helps explain why our full method outperforms the variant without reweighting.
> > >
> > > **Clear set statistics on the long-tailed CIFAR-100**
> > >
> > > | **Methods**	                | **Majority-Count** | **Minority-Count** |**Majority-Acc** | **Minority-Acc** |
> > > |---------|----------|----------|----------|----------|
> > > | **Ours** 		      | 5693 |153 | 78.54% | 83.01% |
> > > | w/o reweighting | 5990  | 140 | 78.75% | 82.14% |
> > >
> > >
> > > **Case 2: The foundation model is accurate and confident only on majority classes.**
> > >
> > > In this case, minority-class samples generally have high model-predicted entropy. As a result, even with the reweighting factor, they are unlikely to be selected into the clean set. On long-tailed PathMNIST, both our method and the variant without reweighting select only majority-class samples into the clean set.
> > >
> > > Overall, these results show that the effect of the reweighting factor depends on pseudo-label reliability. When minority pseudo-labels are already reliable, the reweighting term can improve minority representation in the clean set. When they are not, it does not automatically admit unreliable minority samples. We will add these results and clarifications to the revised manuscript.

---

### Official Review · Reviewer_QJLm · 2026-03-13

**Soundness:** 3
**Presentation:** 4
**Significance:** 2
**Originality:** 4
**Overall Recommendation:** 5
**Confidence:** 5

**Summary:**

The authors provide a new active learning strategy that achieves an amazing gain of 50% less annotation compared to the best available AL algorithms.
Using a Foundation Model (FM), the authors derive priors for labels of the unlabeled data samples. Using Product of Experts, they derive a measure of agreement between the FM and the small model they are training for the unlabeled data samples. They then use this Product of Expert to derive uncertainty measurements for each data samples. Given a predetermined percentage (p), they will then choose the top-p samples with highest uncertainty as the noisy set and they will determine the rest of the set as clean set. They will use the FM determined labels for this clean set as the labels.
For the noisy set, they then calculate the predictive uncertainty of the small model to choose the top-k to be sent to oracle for labels. This way they use the power of PoE to pseudo label a percentage of the samples where they are fairly confident about. this in turn will reduce the labelling need by smart use of PoE.
the authors also assume a weight proportional to the number of samples in a class (based on FM's label decision) to further augment their PoE derived uncertainty calculation to be class imbalanced aware. This way they will make sure to take into account the class imbalance which will effect the labeling budget based on the fairly thorough ablation study.
The authors also show that their method is performing the best among the existing top AL algorithms when further labeling noise added to the equation of already complex class imbalance scenario.
With that the authors show the superiority of the algorithm which is tested both in Text and Image domain for multi-class classification problem.

**Compliance With Llm Reviewing Policy:**

Affirmed.

**Final Justification:**

After reviewing the the newly implemented experiments, I am satisfied with the paper value. Thank you

**Key Questions For Authors:**

No further questions, except the concerns I mentioned above.

**Limitations:**

yes

**Strengths And Weaknesses:**

## Strengths
- The paper is very well-written. The authors took time to explain their problem and their approach and explain the novelty of the work. The algorithm is very well explained and easy to follow and generally the approach is sound and reasonable.
- I specifically like the experiments being done in both text and image domain, they have compared their algorithm with the top best existing AL algorithms and produced a comprehensive result that clearly shows the advantage of their work in reducing required annotation.
- I really like the smart use of PoE to bring in the FM prior into calculation and further use that generate confident pseudo-labels.
- The ablation study to see the effect of class imbalance weight and the label noise is great.

## Weaknesses
- If this paper was produced back in 2023, I would have definitely considered it as a very strong paper. However, at this time, I cannot help but ask if we use an FM model for prior and help in pseudo-labeling, then why would we even need to train a small model? For me to be able to accept the paper as a major contribution, I at least need to see that the trained small model performs better than the FM model used for prior. I would strongly suggest to the authors that add this experiment to their paper. If they already have the balanced accuracy results, running the FM in inference on their heldout test set and reporting the same metric on FM model's result should not take much but will make a convincing conviction about the value of the work.

- Time cost. One of the main concerns about AL algorithms is the time spent in training the model. One weakness that I see is that there is no comparison between the training time between their method versus different reported methods. I can only assume that their method should work faster than BADGE for example but it would have been nice to see the numbers.

- Lastly, I see an ablation study on the percentage of accepted pseudo-labels in each iteration having the same oracle annotation budget would have been nice to further validate if the accuracy increases or decreases by incorporating less pseudo labels in the training set (i.e. the clean set size)

All in all, I really like the method and my only concern is to show the value in the age of generative AI. To show the value, I would like to see all this effort make sense and we end up having a significantly better small model vs the FM. If not, even though the algorithm is brilliant, I am failing to see the value of training a model that cannot match the prior FM.

---

> ### Author Rebuttal · Authors · 2026-03-30
>
> Dear reviewer,
>
> Thank you for taking the time to review our paper and provide thoughtful comments. We have outlined our responses and clarifications below. We hope that our efforts have addressed your concerns, and we kindly request that you consider increasing your scores.
>
> **Response to Weakness 1:** Thank you for this insightful comment. We agree that this comparison is important for clarifying the value of training the small model. To directly address this point, we evaluate the same foundation model used for prior generation on the held-out test set and report its balanced pool accuracy and balanced test accuracy alongside that of the trained small model. For consistency with the main experiments, we report results under the same data setting and average over the same four seeds.
>
> | Model | Balanced Pool Accuracy | Balanced Test Accuracy |
> |:--|:--:|:--:|
> | FM | 50.10% | 50.06% |
> | Small Model | **64.16%** | **90.35%** |
>
> As shown above, in this challenging setting with class imbalance and label noise (noisy PathMNIST with merge imbalance), the trained small model substantially outperforms the foundation model itself on balanced test accuracy. This result suggests that training the small model is not redundant: our framework does not simply copy the FM predictions, *but instead leverages and exploits FM priors to train a downstream model that can **match or surpass** the FM in performance.* We believe this result is important because it demonstrates the practical value of PriorAL. Even when the FM is imperfect, its prior information can still be effectively exploited to train a stronger and more task-adapted small model.
>
> **Response to Weakness 2:** Thank you for raising this important point. We agree that computational cost is an important practical consideration for active learning methods. To address this concern, we rerun the experiments on the noiseless Trec dataset with merge imbalance and compare the runtime of PriorAL and BADGE under the same setting. Since the two major computational components are data selection and model training, we report them separately for clarity. All results are averaged over four random seeds.
>
> | Method     | Data Selection Time (s) | Training Time (s) |
> |------------|--------------------------|-------------------|
> | BADGE      |             221.40           |       3850.33           |
> | PriorAL (Ours) |            176.68              |        3965.94         |
>
> Both methods are run for 20 active learning iterations. As shown above, PriorAL requires **less data selection time** than BADGE, indicating that its selection procedure is relatively efficient. Its **training time is slightly higher**. We attribute this additional training cost mainly to the larger effective training pool used by PriorAL, whereas BADGE trains only on the actively selected labeled subset. PriorAL achieves lower selection cost, while its slightly higher training time reflects the trade-off of leveraging additional pseudo labeled data for improved performance.
>
> **Response to Weakness 3:** Thank you for this thoughtful comment. We agree that an ablation on the proportion of accepted pseudo labels (size of clean set) under the same oracle annotation budget is important for understanding whether incorporating more pseudo labeled samples helps or hurts performance.
> To address this, we conduct additional experiments on the long-tailed Trec dataset without label noise, where we vary the accepted pseudo label proportion while keeping the oracle annotation budget fixed. We then report the balanced pool accuracy and balanced test accuracy at 20% annotation cost, averaged over four random seeds.
>
> | Size of clean set  | Balanced Pool Acc (20%) | Balanced Test Acc (20%) |
> |---|---|---|
> | 21 | **96.20%** |**94.25%** |
> | 210 | 82.07% | 79.45%|
> | 420 | 77.35% | 76.53% |
>
> The results show a clear trend in this setting: accepting more pseudo labeled samples leads to worse performance under the same oracle annotation budget. This suggests that, *when the foundation model (FM) prior is less reliable, increasing the clean set size may introduce enough noisy supervision to outweigh the benefit of having more pseudo labeled data.* Therefore, this ablation supports our design choice that the clean-set size should not be made as large as possible by default. Instead, it should be adapted to the **pseudo label quality of the FM**. In the long-tailed Trec setting, a smaller clean set is more effective, whereas in settings where FM pseudo labels are more reliable, a larger clean set can be more beneficial.

---

> > ### Author Rebuttal · Reviewer_QJLm · 2026-04-06
> >
> > Thank you so much for addressing all the questions. Please add the newly implemented experiments to the paper. At this point I dont have any other concerns and the paper is an accept in my opinion. Thank you

---

> > > ### Author Response · Authors · 2026-04-06
> > >
> > > Thank you very much for your kind and encouraging feedback. We are glad that our responses have addressed your concerns. We will incorporate the newly implemented experiments into the revised version of the paper to further strengthen the presentation.

---

### Official Review · Reviewer_ThLT · 2026-03-24

**Soundness:** 3
**Presentation:** 3
**Significance:** 2
**Originality:** 2
**Overall Recommendation:** 3
**Confidence:** 5

**Summary:**

This paper aims to improve active learning under class imbalance and label noise by leveraging a foundation model. The method combines predictions from a pretrained foundation model and a smaller target model through multiplication, and computes entropy on top of this joint prediction. It further incorporates pseudo label based class weights to address imbalance. Based on the resulting entropy, a subset of data is identified as a “noise set” using a predefined ratio ρ, and sampling is performed within this set using the target model’s entropy. The remaining data are treated as clean and assigned pseudo labels. The approach is evaluated on several image datasets (CIFAR10, CIFAR100, PathMNIST) and text datasets (TREC, AGNews).

**Compliance With Llm Reviewing Policy:**

Affirmed.

**Final Justification:**

I still have concerns about the reliance on pseudo-labeling and its implications for active learning. However, the additional experiments (SSL baseline, structured noise) partially address my empirical concerns. Based on these clarifications, I raised my score to weak reject.

**Key Questions For Authors:**

See weaknesses

**Strengths And Weaknesses:**

## **Paper Strengths**

- The paper is clearly written and easy to follow.
- The problem setting, addressing both class imbalance and label noise, is practically relevant.

## **Paper Weaknesses**

### Limited novelty and technical contribution

- Related directions have already explored: modifying AL queries with foundation model predictions [a], leveraging confident labels for distribution estimation [b, c], and incorporating foundation models into the learning pipeline.
- The use of product-of-experts is a simple combination rather than a fundamentally new design.
- Entropy reweighting and class balancing are largely heuristic.
- The clean versus noisy split is based on thresholding, without a principled formulation.

### Issues in experimental setup

- Comparisons are not fully fair, as it is unclear how foundation models are used in the baselines.
- If the method implicitly benefits from semi-supervised effects, relevant semi-supervied learning baselines should be included.
- Only standard random label flipping noise is considered, which does not reflect realistic noise patterns. Evaluations under structured or biased noise would be more informative, especially for active learning.

### Limited empirical gains

- The method does not consistently achieve state-of-the-art performance across benchmarks.
- In several cases, the improvement over foundation model baselines is marginal.

[a] Bernhardt, Mélanie, et al. "Active label cleaning for improved dataset quality under resource constraints." *Nature communications* 13.1 (2022): 1161.

[b] Hwang, Sehyun, et al. "Combating label distribution shift for active domain adaptation." European Conference on Computer Vision. Cham: Springer Nature Switzerland, 2022.

[c] Cai, Lile, et al. "Revisiting superpixels for active learning in semantic segmentation with realistic annotation costs." *Proceedings of the IEEE/CVF conference on computer vision and pattern recognition*. 2021.

---

> ### Author Rebuttal · Authors · 2026-03-30
>
> Dear reviewer,
>
> We sincerely thank the reviewer for the valuable feedback and constructive suggestions. Below, we provide our responses and clarifications. We hope these help address the reviewer’s concerns.
>
> **Response to Weakness 1:** Thank you for pointing out these related directions. We agree that [a–c] are relevant and will clarify them in the paper. However, they do not address our main setting: active learning under noisy annotations, class imbalance, cross-modal data, and imperfect foundation model priors. Specifically, [a] studies **image-domain** active labeling under label noise **without foundation model priors** and assumes **true labels**; [b] uses confident pseudo labels only for global label-distribution estimation in **active domain adaptation**, unlike our instance-level PoE integration; and [c] studies region-based active learning for semantic segmentation, **without foundation models**, **noisy annotated labels**, or our cross-modal setting.
>
> **Response to Weakness 2:** Thank you for this thoughtful comment. We agree that PoE itself is not novel and do not claim novelty in the operator alone. Our contribution is to use PoE in a noisy, imbalanced, and cross-modal active learning pipeline, where it provides a **reliability signal** for clean/noisy partitioning and helps exploit the *complementarity between foundation model priors and the small model*.
>
> **Response to Weakness 3:** Thank you for this comment. We acknowledge that entropy reweighting and class balancing are heuristic rather than derived from a unified theoretical objective. However, they are introduced to address noisy and imbalanced active learning, where standard uncertainty measures may be biased or less reliable. Empirically, these components remain helpful across modalities, imbalance types, and both noisy and noiseless settings.
>
> **Response to Weakness 4:** Thanks for pointing this out. We agree that the clean/noisy split is heuristic rather than formally derived. However, the threshold is not arbitrary: under the same long-tailed Trec setup as Table 1, the clean set shows **74.46%** higher ground-truth pseudo label accuracy than the noisy set, suggesting that the *split captures reliability*. A more principled or adaptive strategy is valuable future work.
>
> **Response to Weakness 5:** Thank you for this comment. We apologize for the unclear description. Our baselines include classic active learning methods, which do not use foundation models and are kept in their standard formulations, and Pretrained Priors baseline, which uses foundation model priors and is therefore the most directly comparable baseline.
>
> **Response to Weakness 6:** Thank you for this insightful comment. We agree that comparison with a semi-supervised baseline is useful, although our setting is not standard semi-supervised learning. On noiseless long-tailed CIFAR-10 under a matched protocol with ResNet-50 and a 20% label budget, PriorAL achieves **51.49%** balanced test accuracy versus **50.48%** for FlexMatch. This suggests that PriorAL’s gains are not solely explained by generic pseudo labeling, and demonstrates that its contribution is distinct from conventional semi-supervised methods.
>
> **Response to Weakness 7:** Thank you for this valuable suggestion. We agree that standard random flipping does not fully capture realistic noise patterns. We therefore add an experiment under structured noise, where some classes are flipped to semantically similar classes (e.g., BIRD → AIRPLANE). On long-tailed CIFAR-10 with 10% structured noise, PriorAL achieves the best minority-class recall (**26.47%**) while remaining competitive in balanced test accuracy (**41.58%**), compared with Pretrained Priors (**16.57%**, **42.38%**), BADGE (**6.87%**, **30.78%**), and GALAXY (**6.93%**, **30.23%**). These results suggest robustness beyond standard random-noise settings.
>
> **Response to Weakness 8:** Thank you for this valuable comment. We agree that PriorAL is not best on every benchmark or metric. Rather than optimizing for a single setting, our goal is to *develop a method that remains effective across diverse challenging scenarios*. **Across 16 cases** spanning modalities, imbalance types, and label-noise conditions, **PriorAL shows consistent and competitive performance**. We therefore view its main value as robustness across complex imbalanced settings.
>
> **Response to Weakness 9:** Thank you for this comment. We agree that gains over the foundation model baseline may be small when the foundation model is already strong. Our goal, however, is to provide a robust framework that can effectively use foundation model priors across different modalities, imbalance types, and label-noise conditions. Importantly, *our results also show that PriorAL is not merely copying these priors: in several challenging cases, the trained small model matches or even surpasses the foundation model baseline.*

---

> > ### Author Rebuttal · Reviewer_ThLT · 2026-04-03
> >
> > I still have concerns about the reliance on pseudo-labeling and its implications for active learning. However, the additional experiments (SSL baseline, structured noise) partially address my empirical concerns. Based on these clarifications, I raised my score to weak reject.

---

> > > ### Author Response · Authors · 2026-04-05
> > >
> > > Thank you very much for your thoughtful follow-up and for taking the time to reconsider your score. We deeply appreciate your continued engagement with our work and your careful reassessment of the paper.
> > >
> > > As combining foundation-model pseudo labels with active learning is one of the main contributions of our paper, we aimed to evaluate this setting as comprehensively as possible. To this end, we included extensive comparisons with both classical active learning baselines and stronger baselines that also make use of pseudo labels together with true labels, including FlexMatch and Pretrained Priors. Together with the additional structured-noise results, we hope these experiments provide further evidence for the effectiveness of our approach and its strong overall performance in noisy and imbalanced settings.
> > >
> > > We are sincerely grateful for your time, thoughtful comments, and constructive feedback.

---

### Decision · Program_Chairs · 2026-04-30

**Decision:**

Accept (regular)

**Comment:**

The overall assessment is mixed, with several concerns remaining unresolved. The most prominent issue is limited novelty, as key components, such as product-of-experts aggregation, pseudo-labeling, and entropy-based selection, are viewed as incremental or heuristic combinations of existing ideas rather than a fundamentally new approach. In addition, there are open questions regarding robustness and reliability, particularly in scenarios where foundation model pseudo-labels are inaccurate, biased, or concentrated on minority classes. While the rebuttal provides additional analysis, it does not fully resolve concerns about minority-class performance and behavior across active learning iterations.

The rebuttal for two reviewers were successful and led to increased ratings.

Reviewer mW7n recommended 5 (Accept), but he/she did not champion the acceptance of the paper. Thus, his/her rating 5 is not entirely reflected in my overall rating.

Overall, I would like to recommend weak accept.